# Deep Continuous-Time State-Space Models for Marked Event Sequences

**Yuxin Chang**[*]
University of California, Irvine

**Alex Boyd**[*]
GE HealthCare

**Cao Xiao**
GE HealthCare

**Taha Kass-Hout**
GE HealthCare

**Parminder Bhatia**
GE HealthCare

**Padhraic Smyth**
University of California, Irvine

**Andrew Warrington**
GE HealthCare

`yuxinc20@uci.edu, smyth@ics.uci.edu`
`{alex.boyd, andrew.warrington}@gehealthcare.com`

## Abstract

Marked temporal point processes (MTPPs) model sequences of events occurring at irregular time intervals, with wide-ranging applications in fields such as healthcare, finance and social networks. We propose the *state-space point process* (S2P2) model, a novel and performant model that leverages techniques derived for modern deep state-space models (SSMs) to overcome limitations of existing MTPP models, while simultaneously imbuing strong inductive biases for continuous-time event sequences that other discrete sequence models (i.e., RNNs, transformers) do not capture. Inspired by the classical linear Hawkes processes, we propose an architecture that interleaves stochastic jump differential equations with nonlinearities to create a highly expressive intensity-based MTPP model, without the need for restrictive parametric assumptions for the intensity. Our approach enables efficient training and inference with a parallel scan, bringing linear complexity and sublinear scaling while retaining expressivity to MTPPs. Empirically, S2P2 achieves state-of-the-art predictive likelihoods across eight real-world datasets, delivering an average improvement of 33% over the best existing approaches.

## 1 Introduction

Marked temporal point processes (MTPPs) are used to model irregular sequences of events in continuous time, where each event has an associated type, often called a *mark*. MTPPs model the joint distribution of these sequences of event times and marks. They have been successfully applied to modeling purchasing patterns in e-commerce [Türkmen et al., 2019a, Vassøy et al., 2019, Yang et al., 2018], patient-specific medical events [Hua et al., 2022], disease propagation [Gajardo and Müller, 2023], and event modeling and prediction across multiple other domains [Williams et al., 2020, Sharma et al., 2018, Wang et al., 2024]. An MTPP can be fully characterized by a *marked intensity process*, specifying the instantaneous rate of occurrence of each mark conditioned on history.

State-of-the-art neural methods compute hidden states to summarize the event history, which are then used to compute marked intensities at any point in time. However, many models are limited by inexpressive temporal dynamics, lack of support for long-range dependencies, and serial computation [Du et al., 2016, Mei and Eisner, 2017]. Recent advances in transformer-based MTPPs improved performance and gained parallelism, but scale quadratically in sequence length [Zhang et al., 2020,

---

[*]Authors contributed equally

39th Conference on Neural Information Processing Systems (NeurIPS 2025).

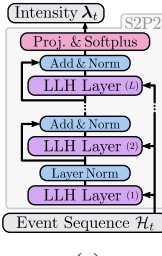

| | | Rank (Lower ↓ is better, ⊠ = best, x = second-best) | | | | | | |
| | | Likelihood | | Next Event Prediction | | Calibration | | Composite |
| Model | | Mark | Time | Mark Acc. | Time RMSE | Mark | Time | Rank |
|---|---|---|---|---|---|---|---|---|
| RMTPP | [Du et al., 2016] | 6.8 | 6.1 | 6.9 | 5.0 | 5.8 | 6.1 | 6.1 |
| SAHP | [Zhang et al., 2020] | 6.9 | 4.9 | 7.1 | 4.3 | 6.8 | 5.4 | 5.5 |
| THP | [Zuo et al., 2020] | 5.5 | 7.0 | 5.6 | 4.3 | 5.0 | 5.4 | 5.5 |
| IFTPP | [Shchur et al., 2020a] | 4.1 | 3.1 | 5.0 | 4.0 | 1.8 | 2.6 | 3.6 |
| MHP | [Gao et al., 2024] | 5.4 | 6.4 | 4.3 | 5.5 | 4.8 | 6.0 | 5.3 |
| NHP | [Mei and Eisner, 2017] | 2.4 | 3.3 | 1.8 | 2.4 | 4.9 | 3.6 | 2.9 |
| AttNHP | Yang et al. [2022] | 2.4 | 3.3 | 2.9 | 6.6 | 3.4 | 3.7 | 3.7 |
| **S2P2** | (Ours) | 1.9 | 1.4 | 1.9 | 2.3 | 3.0 | 2.8 | 2.1 |

(a)            (b)

Figure 1: **(a)** A schematic of our proposed *state-space point process* (S2P2), a deep stack of novel *latent linear Hawkes* (LLH) layers interleaved with nonlinear and normalization layers creating an expressive MTPP architecture. **(b)** Summary table of results we present in Section 6. We summarize ranks for six key metrics, ranking the average held-out test set performance across eight real-world datasets for five randomly seeded models; as well as a holistic *composite rank*, defined as the average of the ranks. Our S2P2 model outperforms all baselines by almost an entire rank, strongly indicating state-of-the-art performance and robustness across metrics and datasets.

Zuo et al., 2020, Yang et al., 2022], preventing them from being used in practice for long sequences, such as modeling clinical events over a patient's entire medical history spanning many years.

Recently, deep state-space models (often abbreviated as SSMs) have emerged as a challenger to transformer-based models for discrete sequence modeling [Gu et al., 2022b, Smith et al., 2022, Gu and Dao, 2023]. SSMs interleave a stack of linear state-space recurrences with position-wise nonlinearities [Gu et al., 2021]. This architecture not only achieves superior performance on a wide range of tasks [Goel et al., 2022, Deng et al., 2023], but retains linear scaling, can be parallelized across the length of a sequence, and can gracefully handle irregularly spaced observations.

Despite being defined in continuous-time, applying SSMs to MTPPs is not straightforward. Tying the conditional intensity to the model's state is the obvious choice, but SSMs expect a continuous-valued signal to integrate as input, whereas event sequences are discontinuous by nature. Taking inspiration from classical parametric linear Hawkes processes (LHPs) [Hawkes, 1971], we introduce a stochastic jump differential equation on the complex plane to serve as the SSM recurrence. We refer to this as a *latent linear Hawkes* (LLH) layer. We derive closed-form update rules for any time that allow the diagonalized continuous-time SSM system to use efficient parallel scans. We then introduce our *state-space point process* (S2P2) model, a stack of LLH layers interleaved with position-wise nonlinear functions (Fig. 1a), inspired by deep SSMs. This design yields an MTPP that has both a highly flexible conditional intensity function and access to efficient, parallelizable computation.

This paper is organized as follows: **In Section 2** we present the necessary preliminaries on MTPPs, LHPs, and deep SSMs. **In Section 3** we introduce our state-space point process architecture. We first build the connection between LHPs and deep SSMs, and then derive how to use this connection to make a highly expressive, parsimonious and parallelizable MTPP. **In Section 5** we then verify this expressivity on small-scale, targeted synthetic explorations comparing to key baselines. **In Section 6** we empirically evaluate our model at scale on a range of metrics across eight real-world datasets, finding that S2P2 matches or exceeds the average predictive performance of baselines, achieving either best- or second-best average performance on all six metrics. These results are summarized in Table 1b. **In Sections 7 and 8** we conclude by discussing the relative advantages and limitations of S2P2 and promising future opportunities. Due to its robustness, performance, and efficiency, our S2P2 model is a powerful model out-of-the-box for a wide range of MTPP applications.

## 2 Preliminaries

### 2.1 Marked Temporal Point Processes

We define an event sequence, or *history*, as a sequence of time-mark pairs, $\mathcal{H}_t := \{(t_i, k_i) \mid t_i \leq t \text{ for } i \in \mathbb{Z}^+\}$, where $t_i \in \mathbb{R}_{\geq 0}$, $\forall i : t_{i-1} < t_i$, and $k_i \in \mathcal{M}$.[2] In this paper, we focus on discrete and

---

[2]Please refer to Tables 3 and 4 in Appendix A for a list of notation and acronym definitions, respectively.

finite mark spaces, i.e., $\mathcal{M} := \{1, \ldots, K\}$; however, $\mathcal{M}$ can be more general, such as countable or continuous. We also define $\mathcal{H}_{t-}$ similarly to $\mathcal{H}_t$, except that it excludes events at exactly time $t$.

One way of characterizing an MTPP is through a *marked intensity process*. The intensity $\boldsymbol{\lambda}_t := [\lambda_t^1, \ldots, \lambda_t^K]^\top \in \mathbb{R}_{\geq 0}^K$ characterizes an MTPP by describing the rate of occurance of events:

$$\lambda_t^k \mathrm{d}t := \mathbb{E}\left[\text{event of type } k \text{ occurs in } [t, t + \mathrm{d}t) \mid \mathcal{H}_{t-}\right] \tag{1}$$

with the total intensity $\lambda_t := \sum_{k=1}^K \lambda_t^k$ being the rate that *any* event occurs. This intensity also defines a marked counting process $\mathbf{N}_t := [N_t^1, \ldots, N_t^K]^\top \in \mathbb{Z}_{\geq 0}^K$, representing the number of occurrences of events of each type of mark in the time span $[0, t]$.

Parameterized forms of $\boldsymbol{\lambda}$ are often trained by optimizing the log-likelihood over observed data (e.g., [Mei and Eisner, 2017, Zuo et al., 2020]). The log-likelihood for a single sequence $\mathcal{H}_T$ is defined as [Daley and Vere-Jones, 2003, ch. 7.3]:

$$\mathcal{L}(\mathcal{H}_T) := \sum_{i=1}^{N_T} \log \lambda_{t_i}^{k_i} - \int_0^T \lambda_s \mathrm{d}s. \tag{2}$$

**Linear Hawkes Processes**   A foundational MTPP is the *linear Hawkes process* (LHP). The LHP is a *self-exciting process*, where event occurrence increases the rate of occurrence of other events, with the influence decaying according to a *kernel*. The intensity function is the summation of influences, and if the kernel is the exponential function, it has the following integral and differential forms:

$$\boldsymbol{\lambda}_t = \boldsymbol{\nu} + \sum_{i=1}^{N_{t-}} \exp\left(-\boldsymbol{\beta}(t - t_i)\right)\boldsymbol{\alpha} \iff \mathrm{d}\boldsymbol{\lambda}_t = -\boldsymbol{\beta}(\boldsymbol{\lambda}_{t-} - \boldsymbol{\nu})\mathrm{d}t + \boldsymbol{\alpha}\mathrm{d}\mathbf{N}_t, \tag{3}$$

where, to ensure non-negative marked intensities, $\boldsymbol{\nu} \in \mathbb{R}_{\geq 0}^K$ and $\boldsymbol{\alpha}, \exp\left(-\boldsymbol{\beta}\right) \in \mathbb{R}_{\geq 0}^{K \times K}$ with $\exp$ being the matrix exponential. This form (which is the most common) of the LHP is incredibly limited, and hence is used primarily for its interpretability, as opposed to outright predictive performance.

## 2.2   Deep State-Space Models

*Deep state-space models* (SSMs) are a class of recurrent models that have excelled in long-range sequence and language modeling tasks, all while having favorable computational properties [Gu et al., 2022b, Smith et al., 2022, Gu and Dao, 2023]. The backbone of deep SSMs are linear state-space models, which define a continuous-time dynamical system with inputs and outputs $\mathbf{u}(t), \mathbf{y}(t) \in \mathbb{R}^H$ through linear differential equations:

$$\frac{\mathrm{d}}{\mathrm{d}t}\mathbf{x}(t) = \mathbf{A}\mathbf{x}(t) + \mathbf{B}\mathbf{u}(t), \quad \mathbf{A} \in \mathbb{R}^{P \times P}, \mathbf{B} \in \mathbb{R}^{P \times H} \tag{4}$$

$$\mathbf{y}(t) = \mathbf{C}\mathbf{x}(t) + \mathbf{D}\mathbf{u}(t), \quad \mathbf{C} \in \mathbb{R}^{H \times P}, \mathbf{D} \in \mathbb{R}^{H \times H} \tag{5}$$

where $\mathbf{x}(t) \in \mathbb{R}^P$ is the (hidden) state of the system, and $\mathbf{A}, \mathbf{B}, \mathbf{C}, \mathbf{D}$ define the system's dynamics. Deep SSMs interleave these recurrences with nonlinear position-wise functions $\sigma$ as $\mathbf{u}^{(l)}(t) := \sigma(\mathbf{y}^{(l-1)}(t))$ (for layer $l$). This yields a sequence model where each recurrence is conditionally linear in time but is *nonlinear* overall due to $\sigma$.

To evaluate an SSM, we first discretize the continuous-time system at appropriate times to yield a discrete sequence of closed-form state updates [Smith et al., 2022]. The resulting discrete-time recurrence can be evaluated using parallel scans [Blelloch, 1990], with linear work scaling and, importantly, sublinear (theoretically logarithmic) scaling of the computation time with respect to sequence length given sufficient parallel compute. This parallel evaluation natively allows varying observation intervals or latent dynamics.

## 3   State-Space Point Process

We seek to define an MTPP model that is (a) highly expressive and (b) can access efficient and parallelizable compute methods. In this section, we formally introduce our *state-space point process* (S2P2) model, as outlined below: **Section 3.1** extends and generalizes the continuous-time form of the LHP, creating a layer we refer to as the *latent linear Hawkes* (LLH) layer. This generalization

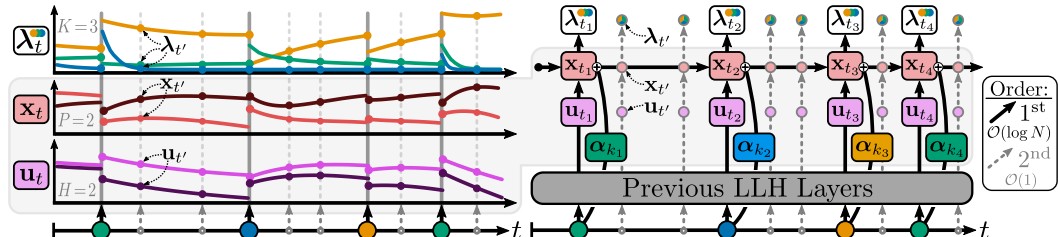

Figure 2: Schematic of our *state-space point process*. We depict the internals of a single LLH layer of the model in their continuous time form (left) and as discrete computations (right). Black arrows can be concurrently computed in logarithmic time, and gray arrows in constant time.

decouples the choice of the width of the layer from the mark space, allowing us to make arbitrarily wide layers, and is the basis for the novel connection between the LHP and deep SSMs. **Section 3.2** discusses how we make each LLH efficient and tractable to compute with parallel scans, and **Section 3.3** extends the model with time-varying dynamics to be more expressive. We conclude in **Section 3.4** by describing S2P2, a highly expressive MTPP model built from the composition of multiple LLH layers and nonlinearities that can be computed with a novel parallel inference scheme.

## 3.1 Continuous-Time Latent Linear Hawkes Layer

To develop a recurrent layer for an MTPP model, we start by reviewing the equations for an LHP intensity, Eq. (3), and an SSM state, Eq. (4). If we allow $\boldsymbol{\nu}$ in the LHP to vary over time, we obtain:

$$\text{LHP:} \qquad d\boldsymbol{\lambda}_t = -\boldsymbol{\beta}\boldsymbol{\lambda}_{t-}dt + \boldsymbol{\beta}\boldsymbol{\nu}_t dt + \boldsymbol{\alpha}d\mathbf{N}_t \quad \in \mathbb{R}^K \qquad (6)$$

$$\text{SSM:} \qquad d\mathbf{x}(t) = \mathbf{A}\mathbf{x}(t)dt + \mathbf{B}\mathbf{u}(t)dt \qquad\qquad \in \mathbb{R}^P. \qquad (7)$$

When presented together the parallels between them become apparent: the LHP intensity, $\boldsymbol{\lambda}_t$ controlled by decay rates $\boldsymbol{\beta}$, is analogous to the state in the linear SSM, $\mathbf{x}(t)$ controlled by state matrix $\mathbf{A}$. The time-varying background LHP intensity, $\boldsymbol{\nu}_t$, is analogous to the SSM input signal after being projected into the state-space, $\mathbf{B}\mathbf{u}(t)$. Compared to SSMs which allow for recurrence over a latent space $\mathbb{R}^P$, Eq. (6) is limited in expressivity due to restricted dynamics $\boldsymbol{\beta}$ and fixed dimensionality tied to the number of possible marks $K$. Not present in Eq. (7) is an impulse $\boldsymbol{\alpha}d\mathbf{N}_t$, which is crucial for allowing the recurrence to condition on abrupt event occurrences over time.

With this in mind, we combine Eqs. (6) and (7) to resolve their respective deficiencies. This results in the following set of stochastic jump differential equations that we call the *latent linear Hawkes* (LLH) layer, which will serve as a continuous-time recurrence layer in our proposed model:

$$\text{LLH:} \qquad d\mathbf{x}_t = -\mathbf{A}\mathbf{x}_{t-}dt + \mathbf{A}\mathbf{B}\mathbf{u}_{t-}dt + \mathbf{E}\boldsymbol{\alpha}d\mathbf{N}_t \quad \in \mathbb{R}^P \qquad (8)$$

$$\mathbf{y}_t = \mathbf{C}\mathbf{x}_t + \mathbf{D}\mathbf{u}_t \qquad\qquad\qquad \in \mathbb{R}^H \qquad (9)$$

where $\boldsymbol{\alpha} \in \mathbb{R}^{R \times K}$, $\mathbf{E} \in \mathbb{R}^{P \times R}$, and the standard SSM parameters share the same dimensionality as in Eqs. (4) and (5). Note that the resulting dynamics within the LLH layer are much more expressive than the LHP that it was inspired by, due to both operating in a separate set of latent dimensions and by having a general dynamics matrix $\mathbf{A} \in \mathbb{R}^{P \times P}$, compared to the more restrictive $\exp(-\boldsymbol{\beta}) \in \mathbb{R}_{\geq 0}^{K \times K}$. For parameter efficiency, we allow the new impulse term $\mathbf{E}\boldsymbol{\alpha}d\mathbf{N}_t$ to be computed as a product of model-wide, low-rank mark embeddings $\boldsymbol{\alpha}$ of rank $R$ and a layer-specific projection into state-space $\mathbf{E}$. For simplicity, we set $R := H$ in practice. Realizations of this layer are shown in Fig. 2.

This LLH will serve as the main form of recurrence in a larger continuous-time model that we define in Section 3.4. In deriving this layer as an extension from LHPs, we retain a strong inductive bias towards event sequences. Furthermore, by drawing connections between this layer and traditional state-space equations, we allow the downstream model to benefit from many of the innovations designed for deep SSMs in recent years, such as parallel computation.

## 3.2 Diagonalizing, Discretizing & Computing the LLH Recurrence

Unlike the LHP intensity, there is no analytical solution to the continuous-time LLH recurrence due to the continuously-integrated $\mathbf{u}(t)$ signal. We must therefore approximate the system at specific

timepoints. If we approximate the input signal by treating it as constant over an update interval, also known as a *zero-order hold* (ZOH) assumption [Iserles, 2009], then we can achieve a closed-form exact update to the recurrence relation. To avoid a computationally expensive matrix exponential in the update rule, we adopt the same general approach as Smith et al. [2022] for deep SSMs by first diagonalizing the system and then imposing the zero-order hold restriction on it. This converts the matrix exponential into an element-wise exponential operation for each LLH layer in S2P2.

**Diagonalization:** Let $-\mathbf{A}$ be diagonalizable with a factorization of $\mathbf{V}\boldsymbol{\Lambda}\mathbf{V}^{-1}$, where $\mathbf{V}, \boldsymbol{\Lambda} \in \mathbb{C}^{P \times P}$ and $\boldsymbol{\Lambda}$ is a diagonal matrix of eigenvalues. An equivalent, diagonalized LLH is then

$$\mathrm{d}\tilde{\mathbf{x}}_t := \boldsymbol{\Lambda}\tilde{\mathbf{x}}_{t-}\mathrm{d}t + \boldsymbol{\Lambda}\tilde{\mathbf{B}}\mathbf{u}_{t-}\mathrm{d}t + \tilde{\mathbf{E}}\boldsymbol{\alpha}\mathrm{d}\mathbf{N}_t \tag{10}$$

$$\mathbf{y}_t := \tilde{\mathbf{C}}\tilde{\mathbf{x}}_t + \mathbf{D}\mathbf{u}_t \tag{11}$$

where $\tilde{\mathbf{x}}_t = \mathbf{V}^{-1}\mathbf{x}_t$, $\tilde{\mathbf{B}} = -\mathbf{V}^{-1}\mathbf{B}$, $\tilde{\mathbf{E}} = \mathbf{V}^{-1}\mathbf{E}$, and $\tilde{\mathbf{C}} = \mathbf{C}\mathbf{V}$. We can then directly parameterize $\tilde{\mathbf{B}}$, $\tilde{\mathbf{C}}$, and $\tilde{\mathbf{E}}$ to avoid learning and inverting $\mathbf{V}$. We use the same initialization strategies as S5 [Smith et al., 2022] for $\tilde{\mathbf{B}}$, $\tilde{\mathbf{C}}$, and $\tilde{\mathbf{E}}$, based off the HiPPO initialization scheme [Gu et al., 2020]. The eigenvalues $\boldsymbol{\Lambda}$ are also directly parameterized and constrained with negative real-components to enforce stability. While the dynamics are diagonalized, we note this *does not* mean that we are modeling the intensities of different mark types as independent. Marks interact through both the position-wise nonlinearities and the learned projections $\tilde{\mathbf{B}}$, $\tilde{\mathbf{C}}$ and $\tilde{\mathbf{E}}$ (since the diagonalized dynamics are equivalent to the original dynamics in Eq. (8), given that the system can be diagonalized on the complex plane).

**Discretization:** We then use a ZOH discretization to create a closed-form update from the diagonalized continuous-time system; please refer to Fig. 2 for an illustration and Appendix B.2 for the full derivation. This results in the following update rule in the diagonalized eigenbasis, that transitions from $\mathbf{x}_t$ to $\mathbf{x}_{t'}$ without a matrix exponential, where, by construction, no events occur in $(t, t')$:

$$\tilde{\mathbf{x}}_{t'} := \begin{cases} \bar{\boldsymbol{\Lambda}}\tilde{\mathbf{x}}_t + (\bar{\boldsymbol{\Lambda}} - \mathbf{I})\tilde{\mathbf{B}}\mathbf{u}_{t'-} & \text{if no event at } t' \\ \bar{\boldsymbol{\Lambda}}\tilde{\mathbf{x}}_t + (\bar{\boldsymbol{\Lambda}} - \mathbf{I})\tilde{\mathbf{B}}\mathbf{u}_{t'-} + \tilde{\mathbf{E}}\boldsymbol{\alpha}_k & \text{if event of type } k \text{ at } t' \end{cases} \tag{12}$$

where $\bar{\boldsymbol{\Lambda}} := \exp(\boldsymbol{\Lambda}(t' - t))$. ZOH is an exact update when $\mathbf{u}$ is constant over the window $[t, t')$; as such, we can choose any value $\mathbf{u}_s$ for $s \in [t, t')$ to hold constant as the input over the integration period. We opt to use $\mathbf{u}_{t'-}$ so that the model can condition on the fact that no events have occurred between $t$ and $t'$. This design decision and its impact on performance are explored in more detail in Appendices B.4 and D.6. It is important that $\mathbf{u}_{t'}$ is not used as the ZOH value to avoid data leakage.

**Computing LLH Recurrence:** The final component is to derive how to use parallel scans to efficiently evaluate the closed-form updates for the modified LLH recurrence in parallel. Parallel scans admit efficient inference over linear recurrences of the form $\mathbf{z}_{i+1} = \mathbf{R}_i\mathbf{z}_i + \mathbf{b}_i$ [Blelloch, 1990]. Although we have an impulse in the recurrence, Eq. (8) is still intrinsically of this form, where $\mathbf{z}_i := \mathbf{x}_{t_i}$, $\mathbf{R}_i := \exp(\boldsymbol{\Lambda}_i(t_{i+1} - t_i))$, and $\mathbf{b}_i := (\mathbf{R}_i - \mathbf{I})\tilde{\mathbf{B}}\mathbf{u}_{t_{i+1}-} + \tilde{\mathbf{E}}\boldsymbol{\alpha}_{k_{i+1}}$. As a result, we can leverage parallel scans to compute the sequence of right-limits $\mathbf{x}_{t_{1:N}}$ in parallel across the sequence length. The corresponding left-limits $\mathbf{x}_{t_{1:N}-}$, which will later be used to calculate event intensities, can then be directly and efficiently computed by subtracting the impulse, $\tilde{\mathbf{E}}\boldsymbol{\alpha}_{k_{1:N}}$, from $\mathbf{x}_{t_{1:N}}$.

### 3.3 Input-Dependent Dynamics

Inspired by recent developments in modern SSMs (e.g., Mamba [Gu and Dao, 2023]), we also consider allowing the dynamics of the system to vary depending on the input and on the history of previous events. This can allow for more expressive intensities. For instance, dynamically adjusting the real components of $\boldsymbol{\Lambda}$ to be smaller will result in more influence from history. Alternatively, larger values will result in more quickly "forgetting" the influence of previous events for a given hidden state channel. This is formalized with the following recurrence relation for $t \in (t_i, t_{i+1}]$:

$$\mathrm{d}\tilde{\mathbf{x}}_t := \boldsymbol{\Lambda}_i\tilde{\mathbf{x}}_{t-}\mathrm{d}t + \boldsymbol{\Lambda}_i\tilde{\mathbf{B}}\mathbf{u}_{t-}\mathrm{d}t + \tilde{\mathbf{E}}\boldsymbol{\alpha}\mathrm{d}\mathbf{N}_t, \tag{13}$$

where $\boldsymbol{\Lambda}_i := \mathrm{diag}\left(\mathrm{softplus}(\mathbf{W}'\mathbf{u}_{t_i} + \mathbf{b}')\right)\boldsymbol{\Lambda}$ with $\mathbf{W}' \in \mathbb{R}^{P \times H}$ and $\mathbf{b}' \in \mathbb{R}^P$. This is conditionally linear in time, as even though $\boldsymbol{\Lambda}_i$ changes, it is entirely input-dependent based on $\mathbf{u}$ and not dependent on previous values of $\mathbf{x}$, and hence we can still use parallel scans as discussed above.

## 3.4 State-Space Point Process Architecture

We are now well-positioned to present the *state-space point process* (S2P2), a flexible and parallel deep continuous-time model for marked intensities $\boldsymbol{\lambda}_t$. This model is shown in Figs. 1a and 2.

We have demonstrated how to define an efficient, parallelizable and scalable core MTPP layer in the LLH layer. While the diagonalized and discretized LLH layer is more expressive than the LHP, it remains fundamentally linear. To compensate, we take inspiration from deep SSMs and alternate $L$ LLH layers with position-wise nonlinearities. This creates a nonlinear model that can still use parallel computation over the sequence length, but is highly expressive compared to each linear recurrence.

We have an input signal in two parts: (i) the continuously-integrated signal $\mathbf{u}_t$ and (ii) the discrete event impulses $\boldsymbol{\alpha}_k$. For the first layer, the only inputs available to condition on are the event impulses themselves, so we set $\mathbf{u}_t^{(1)} = \mathbf{0}$ for all $t \geq 0$. At deeper layers, we have a layer-specific impulse as well as the continuously integrated signal from the previous layer. In general, a layer's output $\mathbf{y}^{(l)} := \text{LLH}^{(l)}(\mathbf{u}^{(l)}, \mathcal{H})$ is passed into a nonlinear activation function $\sigma$ (we use $\sigma(z) := \text{GELU}(z)$ [Hendrycks and Gimpel, 2016]), summed with the residual stream $\mathbf{u}^{(l)}$, and normalized with LayerNorm [Ba, 2016] for the next layer's input. Formally, for $t \geq 0$ and $l = 1, \ldots, L$, then

$$\mathbf{u}_t^{(l+1)} := \text{LayerNorm}^{(l)}(\sigma(\mathbf{y}_t^{(l)}) + \mathbf{u}_t^{(l)}). \tag{14}$$

Due to the unrestricted nature of the recurrences and nonlinearities (and unlike the original LHP), we enforce non-negative intensities by applying an affine projection followed by a rectifying transformation, similar to Mei and Eisner [2017]. This is referred to as the "Proj. & Softplus" layer in Fig. 1, and expressed as $\boldsymbol{\lambda}_t := \mathbf{s} \odot \text{softplus}((\mathbf{W}\mathbf{u}_{t_-}^{(L+1)} + \mathbf{b}) \odot \mathbf{s}^{-1})$ for $t \geq 0$ and where $\odot$ is an element-wise product, $\mathbf{W} \in \mathbb{R}^{K \times H}$, and $\mathbf{b}, \log(\mathbf{s}) \in \mathbb{R}^K$. Then S2P2 can be trained by maximizing the log-likelihood of each sequence, Eq. (2). Similar to other neural MTPPs, we use Monte Carlo estimation for the integral term $\int_0^T \lambda_s dN_s$ [Mei and Eisner, 2017]. Training the model requires computing intensities at both event times $t_{1:N}$ and at randomly sampled times $t \sim \mathcal{U}(0, T)$.

To compute intermediate intensities at these sampled points, we take advantage of the continuous-time nature of S2P2 and *partially evolve* the latent state through the system dynamics. To do so, we first compute the right limits of the hidden states at event times, $\mathbf{x}_{t_i}^{(1:L)}$, as described above (we can do this conditioning in logarithmic depth). Then, Eq. (12) is applied, with no impulse, as no events are occurring at these intermediate points, to find $\mathbf{x}_t^{(1:L)}$. From there, the evolved hidden states are used to evaluate the model across depth to compute the intermediate intensity $\boldsymbol{\lambda}_t$. This operation is both efficient *and* can be done in constant complexity because intermediate evaluations are conditionally independent given the right limits at events. Crucially, there is no separate parametric decoding head, unlike, for instance, Mamba Hawkes processes [Gao et al., 2024] or transformer Hawkes processes [Zuo et al., 2020]. Instead, this tying of the intensity to the model's continuously evolving hidden states, more like the neural Hawkes processes [Mei and Eisner, 2017], makes S2P2 a continuous-time model and contributes to its enhanced expressivity (see Section 5 and Section 7 for more discussions). In Algorithms 1 to 3, we explicitly detail how to use a parallel scan to compute the sequence of right limits at events, how to then evolve those to compute left limits, and then how to subsequently compute the log-likelihood of the sequence.

## 4 Related Work

**Neural MTPPs:** MTPPs are generative models that jointly model the time and type of continuous-time sequential events, typically characterized by mark-specific intensity functions [Daley and Vere-Jones, 2003]. Early approaches used parametric intensity functions, such as self-exciting Hawkes processes [Hawkes, 1971, Liniger, 2009]. More recently, neural models such as RNNs [Du et al., 2016, Mei and Eisner, 2017] and transformers [Zhang et al., 2020, Zuo et al., 2020, Yang et al., 2022] were developed to enable flexible modeling of conditional intensities. Intensity-free MTPPs include normalizing flows [Shchur et al., 2020a, Zagatti et al., 2024], transformers [Draxler et al., 2025], neural processes [Bae et al., 2023], and diffusion models [Zeng et al., 2024, Zhang et al., 2024]; however, modeling intensities is more common, requiring fewer modeling restrictions.

**Efficient MTPPs:** Due to their recurrent nature, RNN-based MTPPs incur $\mathcal{O}(N)$ scaling for length-$N$ sequences as events are processed sequentially. Attention-based MTPP models can be applied in

parallel across the sequence, but with $\mathcal{O}(N^2)$ computational work. Türkmen et al. [2019b] model events as conditionally independent if they occur within the same time bin of a specified size, resulting in parallel computation within bins, but still scaling as $\mathcal{O}(N)$ overall. Shchur et al. [2020b] proposed an intensity-free TPP using triangular maps and the time-change theorem [Daley and Vere-Jones, 2003]. This was extended by Zagatti et al. [2024] to handle marks but losing benefits of the original model and scaling linearly in the mark dimension—which can rapidly become untenable as $\mathcal{O}(NK)$ work. Our S2P2 scales as $\mathcal{O}(\log N)$ and efficiently in marks; more discussion in Appendix B.5.

**SSMs for Sequential Modeling:** SSMs have found recent success as alternatives to RNNs, CNNs, and transformers, enjoying reduced training cost and comparable modelling power [Gu et al., 2022b]. A range of variants have been developed [Gu et al., 2021, Gupta et al., 2022, Gu et al., 2022a, Smith et al., 2022] and applied in language modeling [Gu and Dao, 2023], speech [Goel et al., 2022], and vision [Wang et al., 2023, Zhu et al., 2024]. The linear recurrence enables parallelism, as well as accessible long contexts that are prohibitive for transformers due to quadratic scaling. However, SSMs have not previously been used as continuous-time models for MTPPs, in part due to the intensity functions having different left and right limits, and the input being a stochastic counting process.

**SSMs for TPPs:** Gao et al. [2024] propose using an SSM as a discrete sequence model, encoding an event sequence into a fixed set of static hidden states, then computing intensities with a separate parametric decoder (similar to transformer Hawkes processes [Zuo et al., 2020], but with the transformer replaced with Mamba [Gu and Dao, 2023]). This is fundamentally different to our continuous-time model, where instead of a separate parametric decoding head, we leverage the continuously evolving latent state at any time $t$ to compute predicted intensities at corresponding times. We empirically compare to their model MHP in Section 6, finding that MHP (i) performs comparably to the THP, and (ii) is comprehensively outperformed by S2P2.

# 5 Synthetic Experiments

We first verify the ability of our S2P2 model to represent known intensity functions through a series of targeted experiments on synthetic data. Recent results from the deep SSM literature [Muca Cirone et al., 2024] show that sufficiently deep SSMs are able to approximate arbitrary continuous functions; their results apply to our S2P2 model, and hence we should be able to (verifiably) recover arbitrary and known intensity functions. To demonstrate this, we showcase recovering ground truth intensities in three different settings: the classical point processes of Hawkes and self-correcting processes; an inhomogeneous Poisson process with a discontinuous intensity function; and finally a marked process with long-range dependencies between marks. Our S2P2 recovers the correct intensity in all cases. We include full results for all baselines and reproducibility details for all the following examples and tasks in Appendix D, including intensity visualizations, where S2P2 works as intended, corrects recovers intensities, and matches or exceeds baseline methods performance.

**Classical Point Processes:** We first apply our model and baseline models to two classical point processes: a Hawkes process and a self-correcting process with intensity functions $\lambda_t = 0.5 + \sum_i 0.5 \exp(t_i - t)$ and $\lambda_t = \exp(t - N_t)$, respectively. Examples of the recovered intensity functions are shown in Fig. 4a. We see that our S2P2 recovers the ground truth intensities nearly perfectly. We also explore applying our model and other baseline models to randomly generated synthetic multivariate Hawkes processes in Appendix D.3, finding that all methods perform comparably.

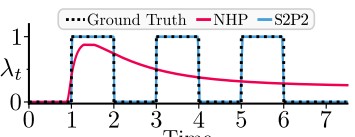

Figure 3: Intensity estimates from trained models when conditioned on an *empty* sequence $\mathcal{H}_t = \emptyset$ for NHP [Mei and Eisner, 2017] and our S2P2. Dotted lines show the ground truth intensity for an inhomogeneous Poisson process. S2P2 accurately captures the background intensity.

**Inhomogeneous Poisson Process:** Our S2P2 does not have a fixed parametric form for the intensity decoder (cf. THP or MHP) or limited recurrent dynamics (cf. NHP). This flexibility should allow the S2P2 to capture intensities where other methods fail. This is showcased in Fig. 3. Models are trained on sequences drawn from an inhomogeneous Poisson process with a square wave for an intensity function (except for $t > 7$). We observe that baseline models fail in predictable ways due to their expressivity limitations (please refer to Appendix D.2 for full results), whereas our model successfully captures the true background intensity process almost perfectly.

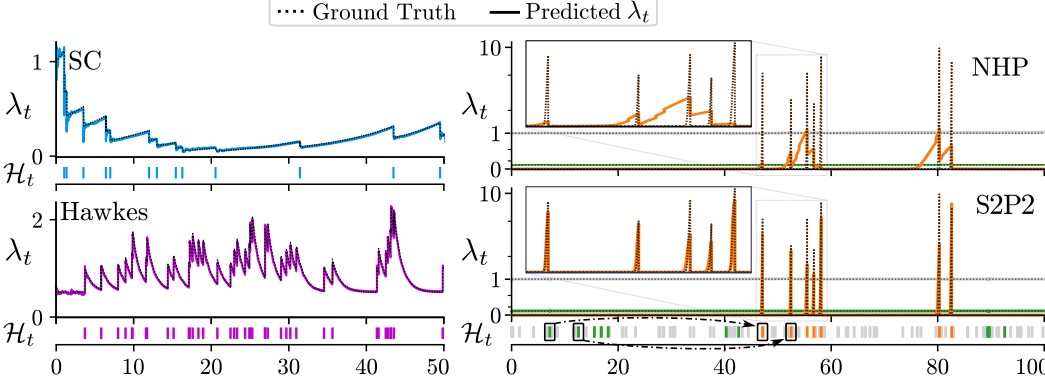

(a) S2P2 successfully recovers parametric intensities from self-correcting and Hawkes processes.

(b) Long-range dependencies between $\mathsf{I} \rightarrow \mathsf{I}$ are accurately captured by S2P2, whereas NHP struggles to recover $\lambda_t$.

Figure 4: Results for synthetic experiments in Section 5.

**Long-Range Dependencies:** Lastly, we study a task with known long-range dependencies, depicted in Fig. 4b. In this task, a "trigger" mark (green) is drawn from a homogeneous Poisson process ($\lambda$=0.1); each trigger is then followed by a "target" mark (orange) a predictably long time later. Here "long" refers to the fact that (a) there are often many trigger marks before a single target mark, and (b) the variance of the distribution over the trigger-target time is comparatively narrow compared to the mean. Gray marks are "distractors" drawn from a homogeneous Poisson process ($\lambda$=1). We see that S2P2 successfully captures the long-range dependencies, whereas NHP struggles. In spite of the long delay between cause and effect within the data, S2P2 was able to recover 98% of the true data likelihood, whereas NHP only achieved 88%. This shows that the state-space point process successfully captures long-range dependencies *and* is flexible as an approximation family.

## 6 Real-World Experiments

We now present the main experiments in this paper, evaluating on real-world datasets.[3] Further results and extensive experimental details are included in Appendices C and D.

**Datasets:** We compare models on eight different datasets, including five datasets available from `EasyTPP` [Xue et al., 2023] (Amazon, Retweet, Taxi, Taobao and StackOverflow). We also add two commonly used datasets in the literature (Last.fm and MIMIC-II), as well as a new medical events dataset derived from the publicly available EHRSHOT dataset [Wornow et al., 2023]. EHRSHOT is an order of magnitude larger than any dataset in `EasyTPP` in both the number of marks and the maximum sequence length (see summary statistics in Table 7), representing a challenging application.

**Metrics:** We performed a comprehensive evaluation using three common families of metrics: (i) total per-event log-likelihood [Mei and Eisner, 2017, Zhang et al., 2020, Zuo et al., 2020, Yang et al., 2022, Gao et al., 2024] (also broken down into mark- and time-specific likelihoods); (ii) predictive accuracy summary metrics of next-mark classification and next event time RMSE [Du et al., 2016, Zhang et al., 2020, Zuo et al., 2020, Yang et al., 2022, Gao et al., 2024]; and (iii) mark and event time calibration metrics [Zhang et al., 2020, Bosser and Taieb, 2023] measuring the reliability of the implied uncertainties (see Appendix D.7). In order to characterize overall performance across multiple diverse metrics, we also compute a summary composite rank in Table 1b, defined as the average rank obtained for a given model across all dataset-metric combinations across five random seeds (i.e., is an average of 240 individual evaluations and five differently initialized trained models).

**Models:** We use the `EasyTPP` library [Xue et al., 2023] for baseline models and S2P2. We perform extensive hyperparameter searches for *each individual pair* of baseline model and dataset, then select

---

[3]Our model is fully integrated into the `EasyTPP` [Xue et al., 2023] library [link]. Other code changes to reproduce our results can be found in our forked repository [link]. Model checkpoints are available on request.

Table 2: Results for S2P2 and baselines across key metrics. We show the mean on the held-out test set evaluated across five random training seeds (std. dev. in parentheses). OOM indicates insufficient memory. We also report the mean rank of models across datasets as a summary metric (for which lower ↓ is better). S2P2 is consistently the best or second best-performing model on average on each dataset and metric. Extended results and discussion are shown in Appendix D.1.

(a) Per event total log-likelihood. Higher values indicate better performance.

| Model | Per Event Log-Likelihood, $\mathcal{L}_{Total}$ (nats) (Higher ↑ is better, **x** = best, x = second-best) | | | | | | | | Average Rank (↓) |
|---|---|---|---|---|---|---|---|---|---|
| | Amazon | Retweet | Taxi | Taobao | StackOverflow | Last.fm | MIMIC-II | EHRSHOT | |
| RMTPP | -2.136 (0.003) | -7.098 (0.217) | 0.346 (0.002) | 1.003 (0.004) | -2.480 (0.019) | -1.780 (0.005) | -0.472 (0.026) | -8.081 (0.025) | 7.1 |
| SAHP | -2.074 (0.029) | -6.708 (0.029) | 0.298 (0.057) | 1.168 (0.028) | -2.341 (0.058) | -1.646 (0.083) | -0.677 (0.072) | -6.804 (0.126) | 5.8 |
| THP | -2.096 (0.002) | -6.659 (0.007) | 0.372 (0.002) | 0.790 (0.002) | -2.338 (0.014) | -1.712 (0.011) | -0.577 (0.011) | -7.208 (0.096) | 6.1 |
| IFTPP | 0.496 (0.002) | -10.344 (0.016) | 0.453 (0.002) | **1.318** (0.017) | -2.233 (0.009) | **-0.492** (0.017) | 0.317 (0.052) | -6.596 (0.240) | 3.0 |
| MHP | -2.091 (0.002) | -6.564 (0.015) | 0.370 (0.008) | 0.636 (0.004) | -2.346 (0.012) | -1.676 (0.004) | -0.351 (0.012) | -7.206 (0.407) | 5.9 |
| NHP | 0.129 (0.012) | **-6.348** (0.000) | 0.514 (0.004) | 1.157 (0.004) | -2.241 (0.002) | -0.574 (0.011) | 0.060 (0.017) | -3.966 (0.058) | 3.0 |
| AttNHP | 0.484 (0.077) | -6.499 (0.028) | 0.493 (0.009) | 1.259 (0.022) | -2.194 (0.016) | -0.592 (0.051) | -0.170 (0.077) | OOM | 3.1 |
| S2P2 (Ours) | **0.781** (0.011) | -6.365 (0.003) | **0.522** (0.004) | 1.304 (0.039) | **-2.163** (0.009) | -0.557 (0.046) | **0.919** (0.069) | **-2.512** (0.369) | **1.4** |

(b) RMSE of the next event time prediction. Lower RMSE values indicate better performance.

| Model | Next Event Time RMSE (Lower ↓ is better, **x** = best, x = second-best) | | | | | | | | Average Rank (↓) |
|---|---|---|---|---|---|---|---|---|---|
| | Amazon | Retweet | Taxi | Taobao | StackOverflow | Last.fm | MIMIC-II | EHRSHOT | |
| RMTPP | 0.338 (0.000) | 16488 (70.5) | 0.283 (0.001) | 0.126 (0.000) | 1.049 (0.000) | 15.873 (0.000) | 0.749 (0.010) | 3425 (12) | 5.0 |
| SAHP | 0.335 (0.001) | 16102 (62.4) | 0.290 (0.008) | 0.126 (0.000) | 1.031 (0.011) | 15.757 (0.007) | 1.142 (0.198) | 3374 (9.4) | 4.3 |
| THP | 0.332 (0.000) | 16268 (18.7) | 0.285 (0.001) | **0.125** (0.000) | 1.033 (0.005) | 15.871 (0.000) | 0.768 (0.005) | 3414 (1.0) | 4.3 |
| IFTPP | **0.327** (0.000) | 16625 (0.2) | 0.362 (0.178) | **0.125** (0.000) | 1.340 (0.724) | 16.508 (0.555) | 0.767 (0.029) | 3616 (17.6) | 5.0 |
| MHP | 0.329 (0.000) | 16109 (36.9) | 0.284 (0.003) | 0.126 (0.000) | 1.046 (0.030) | 15.871 (0.000) | 0.758 (0.065) | 3418 (5.8) | 4.0 |
| NHP | 0.339 (0.000) | **15911** (4.0) | 0.282 (0.001) | 0.126 (0.000) | 1.019 (0.001) | 15.733 (0.008) | **0.726** (0.001) | **3330** (30.9) | 2.4 |
| AttNHP | 2.656 (1.950) | 16171 (284.2) | 1.739 (0.422) | 0.130 (0.000) | 1.256 (0.030) | 15.865 (0.017) | 0.860 (0.022) | OOM | 6.6 |
| S2P2 (Ours) | **0.327** (0.000) | 15987 (13.7) | **0.281** (0.000) | 0.126 (0.000) | **1.014** (0.001) | **15.720** (0.000) | 0.894 (0.054) | 3368 (14.4) | **2.3** |

(c) Next mark classification accuracy (top-10 accuracy on EHRSHOT). Higher values indicate better performance.

| Model | Next Mark Classification Accuracy (%) (Higher ↑ is better, **x** = best, x = second-best) | | | | | | | | Average Rank (↓) |
|---|---|---|---|---|---|---|---|---|---|
| | Amazon | Retweet | Taxi | Taobao | StackOverflow | Last.fm | MIMIC-II | EHRSHOT | |
| RMTPP | 30.8 (0.1) | 53.4 (0.6) | 91.4 (0.1) | 60.9 (0.1) | 45.6 (0.3) | 52.5 (0.1) | 92.3 (0.3) | 36.5 (0.2) | 6.9 |
| SAHP | 32.4 (1.0) | 57.5 (2.2) | 91.4 (0.7) | 60.5 (0.2) | 44.7 (2.0) | 51.8 (0.7) | 86.8 (0.9) | 49.6 (3.1) | 7.1 |
| THP | 34.6 (0.1) | 60.2 (0.1) | 91.4 (0.0) | 60.0 (0.0) | 46.6 (0.2) | 53.3 (0.1) | 90.9 (0.2) | 49.7 (1.7) | 5.6 |
| IFTPP | 35.9 (0.1) | 50.4 (2.5) | 91.8 (0.0) | 61.0 (0.1) | 45.6 (0.1) | 56.4 (0.1) | 93.4 (0.1) | 62.7 (3.8) | 4.3 |
| MHP | 35.1 (0.1) | 60.0 (0.2) | 91.4 (0.3) | 60.7 (0.2) | 46.5 (0.1) | 54.3 (0.1) | 93.2 (0.3) | 54.8 (4.5) | 5.0 |
| NHP | 39.4 (0.1) | **61.4** (0.0) | 92.9 (0.1) | **61.5** (0.2) | 47.1 (0.1) | **56.5** (0.1) | 94.3 (0.0) | 76.5 (0.2) | **1.8** |
| AttNHP | 38.9 (0.9) | 60.7 (0.2) | 92.6 (0.1) | 61.3 (0.2) | **48.2** (0.2) | 55.8 (0.6) | 92.9 (0.6) | OOM | 2.9 |
| S2P2 (Ours) | **40.7** (0.0) | 61.3 (0.0) | **93.1** (0.1) | 61.1 (0.1) | 47.5 (0.3) | 55.8 (0.4) | **96.0** (0.4) | **79.5** (0.3) | 1.9 |

the configurations that maximizes validation log-likelihood per model/dataset. The range of grid search and other details are elaborated in Appendix C.1; results are reported on a fully held-out test set. All models were trained on a single 24GB NVIDIA A5000 GPU.

**Main Results:** We report full numerical results in Table 2, and a summary of rankings in Table 1b. Table 2a shows that S2P2 consistently achieves the best or the second-best held-out log-likelihood results across all datasets, beating other methods by over a whole rank and with a (geometric) mean likelihood ratio of 1.33 (corresponding to a 33% higher likelihood of true events). This is calculated by computing the mean log-likelihood ratio across all datasets and then exponentiating. We further investigate the log-likelihood improvement of S2P2 by separating the time- and mark-specific log-likelihood, finding that S2P2 is mainly driven by better temporal modeling and while achieving gains in both time and mark modeling over existing methods (see Table 8 and Fig. 6 in Appendix D.1 for full results). Additionally, we report prediction summary metrics, next event time prediction and next mark classification, in Table 2b and Table 2c. We again see that S2P2 performs well, matching or exceeding the best-performing baseline, and far outpacing other baseline methods.

**Additional Experiments:** We defer several additional experiments and explorations to Appendices:

1. **Calibration results:** We also include full calibration results, both in raw numerical form and as reliability graphs [Bosser and Taieb, 2023]. These show that S2P2 is well calibrated, placing highest amongst intensity-based methods, only being beaten by IFTPP [Shchur et al., 2020a].
2. **Computational scaling:** We define the theoretical time complexity of each baseline, where S2P2 is best with logarithmic complexity and linear work. We verify this scaling empirically.
3. **Ablations**: We also perform an ablation study on whether input dependent dynamics and the choice of which input to hold constant over the integration interval (introduced in Sections 3.2 and 3.3)

affect performance. We find input-dependent dynamics nearly always improve performance, and that performance is less sensitive to the specific form of the LLH's input signal $\mathbf{u}_t$ in ZOH.

# 7   Discussion

**Continuous-Time Hidden States:**   The S2P2 and NHP [Mei and Eisner, 2017] architectures are closely related, in that both model *continuous-time latent states*. For the NHP, the intensity function is parameterized by a hidden state $\mathbf{h}(t)$ evolved via a continuous-time LSTM variant. Similarly, S2P2 continuously evolves a set of continuous-time latent states $\mathbf{h}^{1:L}(t)$, one for each layer $\{1, \ldots, L\}$. Then the hidden state $\mathbf{h}^L(t)$ of the top layer is decoded into intensities at any $t$. In contrast, models such as THP, MHP, IFTPP, only have latent states $\mathbf{h}(t_j)$ at *discrete event times* $t_{j \in \{1:N\}}$. Consequently, some parametric shapes need to be defined to "interpolate" the hidden state values between events, thus sacrificing models' flexibility due to these parametric functions.

Our results are grouped according to whether or not models have continuous-time hidden states. We empirically verify that *continuously evolving hidden states* enable more expressive latent dynamics, where the expressiveness in explaining the data is measured through log-likelihood; see Appendix D.1. S2P2 further improves scalability by incorporating SSMs that are naturally in continuous-time and by leveraging efficient parallelized scans. S2P2 also introduces a stronger inductive bias than most MTPP models (similar to NHP), helping to achieve superior next event predictions.

**Limitation and Future Directions:**   S2P2 forgoes the interpretability of the parameters and latent states of linear Hawkes processes (LHPs), meaning it is not appropriate when the interpretation of the underlying system is crucial. This tradeoff is common with neural MTPP models (e.g., NHP), because the limited expressivity of LHP may lead to poor predictive performance, as well as misleading interpretations due to underfitting. Therefore, an exciting opportunity for future research is exploiting the connection to recover LHP-like interpretability while retaining the enhanced predictive power of S2P2. For example, any recently developed techniques for interpreting deep SSMs can be extended to S2P2, such as Ali et al. [2025], while fully taking advantage of the continuously varying attention from continuous-time latent dynamics of S2P2 will provide richer insights compared to discrete-time attention maps (e.g., large language models).

Beyond this, further directions include leveraging theories and best practices from deep SSMs, such as the enhanced parameterizations presented by Merrill et al. [2024]. S2P2 can also naturally be extended and accommodate non-categorical marks by tailoring its architecture to the structure of the mark space, such as marks having continuous values or containing richer information. A concrete example is spatio-temporal point processes that have applications in seismic and weather forecasting. Other valuable extensions include serving S2P2 as a pre-trained backbone for downstream applications, such as EHRSHOT clinical classification tasks. Additionally, censoring or adversarial conditions are commonly seen in practice [Boyd et al., 2023, Chakraborty et al., 2025]. Extending the evaluation of S2P2 and MTPP models in general remains an open direction for real-world MTPP deployments.

# 8   Conclusion

In this paper, we present the *state-space point process* (S2P2) model—a novel fusion of concepts from LHPs and SSMs. S2P2 uses deep stacks of stochastic jump differential equations to create an expressive and parsimonious MTPP without additional and restrictive intensity decoding heads, while simultaneously being able to leverage parameterizations and techniques borrowed from deep SSM architectures. We demonstrated that S2P2 outperforms existing methods across a range of standard and new benchmark tasks, and over a range of predictive metrics and efficiency evaluations.

As an intensity-based model, S2P2 requires numerical integration for predicting the expected next event times (as with any neural intensity-based MTPP model) and is not able to model point masses in time (due to the MTPP assumption of no concurrent events). All other baselines suffer from at least one key additional deficiency (e.g., NHP has linear scaling, THP has quadratic work, IFTPP has a parametric decoder, etc.), all of which are resolved by S2P2, while also achieving better performance. We provide a PyTorch implementation via `EasyTPP` for out-of-the-box usage. We believe the robustness, state-of-the-art predictive performance across a range of metrics, computational efficiency, and extensibility of S2P2 make it a very competitive model for a wide range of MTPP applications.

## Acknowledgements

We thank the reviewers for their invaluable feedback on improving the paper. This work was supported by the Hasso Plattner Institute (HPI) Research Center in Machine Learning and Data Science at the University of California, Irvine, by National Science Foundation awards 2505006 and 2425932, and by the National Institutes of Health under awards R01-LM013344 and R01CA297869, and by GE HealthCare.

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

## Supplementary Materials: Deep Continuous-Time State-Space Models for Marked Event Sequences

**Table of Contents**

# A Acronyms and Notation

Table 3: Key notation used repeatedly across this paper.

| Symbol | Space | Description |
|---|---|---|
| $t$ | $\mathbb{R}_{\geq 0}$ | Time |
| $T$ | $\mathbb{R}_{\geq 0}$ | Maximum time in a given sequence's observation window |
| $t_i$ | $\mathbb{R}_{\geq 0}$ | $i^{\text{th}}$ time |
| $t-$ | $\mathbb{R}_{\geq 0}$ | Subscript minus indicates left-limit |
| $t+$ | $\mathbb{R}_{\geq 0}$ | Subscript plus indicates right-limit |
| $k$ | $\mathcal{M} = \{1, \ldots, K\}$ | Event mark |
| $\mathcal{H}$ | $\mathcal{M}^N \times \mathbb{R}_{\geq 0}^N$ | Event history for $N$ events |
| $\mathbf{N}_t$ | $\mathbb{Z}_{\geq 0}^K$ | Counting process for $K$ marks at time $t$ |
| $\lambda_t^k$ | $\mathbb{R}_{\geq 0}$ | Intensity of $k^{\text{th}}$ mark type at time $t$ |
| $\boldsymbol{\lambda}_t$ | $\mathbb{R}_{\geq 0}^K$ | Vector of $K$ mark intensities at time $t$ |
| $\lambda_t$ | $\mathbb{R}_{\geq 0}$ | Ground/total intensity (sum of mark-specific intensities) |
| $\mathcal{L}(\cdot)$ | $\mathbb{R}$ | Log-likelihood of the argument under the model |
| $\nu^{\text{k}}$ | $\mathbb{R}_{\geq 0}$ | Background intensity for the $k^{\text{th}}$ mark |
| $\boldsymbol{\alpha}$ | $\mathbb{R}_{\geq 0}^{K,K}$ | (For LHP) Matrix of intensity impulses from each type of mark |
| $\boldsymbol{\beta}$ | $\mathbb{R}_{\geq 0}^{K,K}$ | (For LHP) Dynamics matrix of intensity evolution |
| $R$ | $\mathbb{Z}^+$ | Mark embedding rank |
| $P$ | $\mathbb{Z}^+$ | LLH/SSM hidden dimension |
| $\mathbf{x}_t$ | $\mathbb{R}^P$ | LLH/SSM hidden state at time $t$ |
| $\mathbf{x}_0$ | $\mathbb{R}^P$ | Learned LLH/SSM initial hidden state |
| $H$ | $\mathbb{N}$ | LLH/SSM output dimension |
| $\mathbf{y}_t$ | $\mathbb{R}^H$ | LLH/SSM output at time $t$ |
| $\mathbf{u}_t$ | $\mathbb{R}^H$ | LLH/SSM input at time $t$ |
| $\mathbf{A}$ | $\mathbb{R}^{P \times P}$ | LLH/SSM transition matrix |
| $\mathbf{B}$ | $\mathbb{R}^{P \times H}$ | LLH/SSM input matrix |
| $\mathbf{C}$ | $\mathbb{R}^{H \times P}$ | LLH/SSM output matrix |
| $\mathbf{D}$ | $\mathbb{R}^{H \times H}$ | LLH/SSM passthrough matrix |
| $\mathbf{E}$ | $\mathbb{R}^{P \times R}$ | LLH mark embedding matrix ($P \times R$ in low-rank factorization) |
| $L$ | $\mathbb{Z}^+$ | Number of linear recurrences in a S2P2 model; model "depth" |
| $\boldsymbol{\alpha}$ | $\mathbb{R}^{R \times K}$ | (For S2P2) Mark impulses ($R \times K$ in low-rank factorization) |
| $\sim$ | N/A | Tilde (e.g., $\tilde{\mathbf{B}}$) denotes variable is in the diagonalized eigenbasis |
| $\Lambda$ | $\mathbb{C}^{P \times P}$ | Matrix of eigenvalues of $\mathbf{A}$; diagonalized dynamics matrix |
| $\bar{\Lambda}$ | $\mathbb{C}^{P \times P}$ | Discretized diagonal dynamics matrix |
| $(l)$ | N/A | Superscript index in parenthesis indicates layer (i.e., $\mathbf{x}$ for layer $l$) |

Table 4: Key acronyms used throughout this paper.

| Acronym | Page number | Definition |
|---|---|---|
| CNN | 7 | Convolutional neural network |
| LHP | 2 | Linear Hawkes process |
| LLH | 2 | Latent linear Hawkes |
| MTPP | 1 | Marked temporal point process |
| RNN | 1 | Recurrent neural network |
| SSM | 1 | (Deep) State-space model |
| TPP | 7 | Temporal point process |
| ZOH | 5 | Zero-order hold |
| RMTPP | 2 | Recurrent marked temporal point process [Du et al., 2016] |
| NHP | 2 | Neural Hawkes process [Mei and Eisner, 2017] |
| SAHP | 2 | Self-attentive Hawkes process [Zhang et al., 2020] |
| THP | 2 | Transformer Hawkes process [Zuo et al., 2020] |
| AttNHP | 2 | Attentive neural Hawkes process [Yang et al., 2022] |
| IFTPP | 2 | Intensity-free temporal point process [Shchur et al., 2020a] |
| MHP | 2 | Mamba Hawkes process [Gao et al., 2024] |
| S2P2 | 1 | State-space point process (ours) |

# B  Additional Details on Methods

## B.1  State-Space Point Process Algorithms

---

**Algorithm 1** State-Space Point Process: Get Right State Limits

---

**Input:** S2P2 layer parameters $\boldsymbol{\Theta} = \left\{ \boldsymbol{\Lambda}^{(l)}, \tilde{\mathbf{B}}^{(l)}, \tilde{\mathbf{C}}^{(l)}, \mathbf{D}^{(l)}, \tilde{\mathbf{E}}^{(l)}, \tilde{\mathbf{x}}_0^{(l)} \right\}_{l=1}^{L}$, event intervals $\Delta t_{1:N}$, nonlinearity $\sigma$, shared mark embeddings $\boldsymbol{\alpha}_{1:N}$.

**Output:** Right state limits $\mathbf{x}_{t_{1:N}}^{(1:L)}$

1: $\mathbf{u}_{t_{1:N}-} = \mathbf{0}$                $\triangleright$ Left input limits

2: **for** $l$ **in** $1 : L$ **do**
3:     $\bar{\boldsymbol{\Lambda}}_{1:N}^{(l)} = \text{Discretize}\left(\boldsymbol{\Lambda}^{(l)}, \Delta t_{1:N}\right)$        $\triangleright$ Zero-order hold, see Eq. (19)
4:     $\tilde{\mathbf{x}}_{t_{1:N}}^{(l)} = \text{ParallelScan}\left(\bar{\boldsymbol{\Lambda}}_{1:N}^{(l)}, (\bar{\boldsymbol{\Lambda}}_{1:N}^{(l)} - \mathbf{I})\tilde{\mathbf{B}}^{(l)}\mathbf{u}_{t_{1:N}-} + \tilde{\mathbf{E}}^{(l)}\boldsymbol{\alpha}_{1:N}\right)$    $\triangleright$ Compute right $x$ limits
5:     $\tilde{\mathbf{x}}_{t_{1:N}-}^{(l)} = \tilde{\mathbf{x}}_{t_{1:N}}^{(l)} - \tilde{\mathbf{E}}^{(l)}\boldsymbol{\alpha}_{1:N}$        $\triangleright$ Compute left $x$ limits
6:     $\mathbf{u}_{t_{1:N}-} = \text{LayerNorm}\left(\sigma\left(\tilde{\mathbf{C}}^{(l)}\tilde{\mathbf{x}}_{t_{1:N}-} + \mathbf{D}^{(l)}\mathbf{u}_{t_{1:N}-}\right) + \mathbf{u}_{t_{1:N}-}\right)$    $\triangleright$ Compute next layer's left $u$ limits
7: **end for**
8: **return** $\mathbf{x}_{t_{1:N}}^{(1:L)}$

---

**Algorithm 2** State-Space Point Process: Get Intensity From Right Limit

---

**Input:** S2P2 layer parameters $\boldsymbol{\Theta} = \left\{ \boldsymbol{\Lambda}^{(l)}, \tilde{\mathbf{B}}^{(l)}, \tilde{\mathbf{C}}^{(l)}, \mathbf{D}^{(l)}, \tilde{\mathbf{E}}^{(l)}, \tilde{\mathbf{x}}_0^{(l)} \right\}_{l=1}^{L}$, Previous state right limits $\mathbf{x}_t^{(1:L)}$, Integration period $\delta t$, nonlinearity $\sigma$, Intensity function IntensityFn.

**Output:** Intensity left limit $\boldsymbol{\lambda}_{t+\delta t}$

1: $\mathbf{u}_{t+\delta t-} = \mathbf{0}$             $\triangleright$ Left input limit

2: **for** $l$ **in** $1 : L$ **do**
3:     $\bar{\boldsymbol{\Lambda}}^{(l)} = \text{Discretize}\left(\boldsymbol{\Lambda}^{(l)}, \delta t\right)$        $\triangleright$ Zero-order hold, see Eq. (19)
4:     $\tilde{\mathbf{x}}_{t+\delta t-}^{(l)} = \bar{\boldsymbol{\Lambda}}^{(l)}\mathbf{x}_t^{(l)} + (\bar{\boldsymbol{\Lambda}}^{(l)} - \mathbf{I})\tilde{\mathbf{B}}^{(l)}\mathbf{u}_{t+\delta t-}$        $\triangleright$ Evolve state
5:     $\mathbf{u}_{t+\delta t-} = \text{LayerNorm}\left(\sigma\left(\tilde{\mathbf{C}}^{(l)}\tilde{\mathbf{x}}_{t+\delta t-}^{(l)} + \mathbf{D}^{(l)}\mathbf{u}_{t+\delta t-}\right) + \mathbf{u}_{t+\delta t-}\right)$    $\triangleright$ Compute event left $u$ limits
6: **end for**

7: $\boldsymbol{\lambda}_{t+\delta t} = \text{IntensityFn}(\mathbf{u}_{t+\delta t-})$        $\triangleright$ Rectify intensity, see Section 3.4
8: **return** $\boldsymbol{\lambda}_{t+\delta t}$

---

**Algorithm 3** State-Space Point Process: Compute Log-Likelihood

---

**Input:** S2P2 layer parameters $\boldsymbol{\Theta} = \left\{ \boldsymbol{\Lambda}^{(l)}, \tilde{\mathbf{B}}^{(l)}, \tilde{\mathbf{C}}^{(l)}, \mathbf{D}^{(l)}, \tilde{\mathbf{E}}^{(l)}, \tilde{\mathbf{x}}_0^{(l)} \right\}_{l=1}^{L}$, Event times $t_{1:N}$, mark types $k_{1:N}$, nonlinearity $\sigma$, shared mark embedding function EmbedMarks, number of integration points per event $M$, Intensity function IntensityFn.

**Output:** Log-likelihood $\mathcal{L}$

1: $\boldsymbol{\alpha}_{1:N} = \text{EmbedMarks}(k_{1:N})$        $\triangleright$ Shared embeddings
2: $t_0 := 0$
3: $\Delta t_{1:N} = t_{1:N} - t_{0:N-1}$
4: $s_{1:N,1:M} \sim \mathcal{U}(0, \Delta t_{1:N})$        $\triangleright$ Sample $M$ integration points per interval (non-inclusive)

5: $\tilde{\mathbf{x}}_{t_{1:N}}^{(1:L)} = \text{GetRightStateLimits}(\boldsymbol{\Theta}, \Delta t_{1:N}, \sigma, \boldsymbol{\alpha}_{1:N})$        $\triangleright$ Algorithm 1, $\mathcal{O}(\log N)$ parallel time

6: **for** $n$ **in** $1 : N$ **do**        $\triangleright$ This is *embarrassingly parallelizable* with vmap, $\mathcal{O}(1)$ parallel time
7:     $\boldsymbol{\lambda}_{t_n} = \text{GetIntensityFromRightLimit}\left(\boldsymbol{\Theta}, \tilde{\mathbf{x}}_{t_n}^{(1:L)}, \Delta t_n, \sigma, \text{IntensityFn}\right)$    $\triangleright$ Algorithm 2, $\mathcal{O}(1)$ parallel time
8:     **for** $m$ **in** $1 : M$ **do**        $\triangleright$ This is *embarrassingly parallelizable* with vmap, $\mathcal{O}(1)$ parallel time
9:        $\boldsymbol{\lambda}_{s_{n,m}} = \text{GetIntensityFromRightLimit}\left(\boldsymbol{\Theta}, \tilde{\mathbf{x}}_{t_n}^{(1:L)}, s_{n,m}, \sigma, \text{IntensityFn}\right)$    $\triangleright$ Algorithm 2, $\mathcal{O}(1)$ parallel time
10:     **end for**
11: **end for**

12: $\mathcal{L} = \sum_{n=1}^{N} \log \lambda_{t_n}^{k_n} + \sum_{n=1}^{N} \frac{\Delta t_n}{M} \sum_{m=1}^{M} \sum_{k=1}^{K} \lambda_{s_{n,m}}^k$        $\triangleright$ Eq. (2) with Monte-Carlo approximation of integral

13: **return** $\mathcal{L}$

---

## B.2 Discretization and Zero Order Hold

The linear recurrence is defined in continuous-time. This mirrors the (M)TPP setting, where event times are not on a fixed intervals. We use the zero-order hold (ZOH) discretization method, to convert the continuous-time linear recurrence into a sequence of closed-form updates, given the integration times, that can also be efficiently computed. We refer the reader to Iserles [2009] for a comprehensive introduction to the ZOH transform.

The main assumption of the ZOH discretization is that the input signal is held constant over the time period being integrated. Under this assumption, it is possible to solve for the dynamics and input matrices that yield the correct state at the end of the integration period. For the LLH dynamics in Eq. (8), when no events occur in $(t, t')$, this becomes

$$\mathbf{x}_{t'-} = \int_t^{t'} \mathbf{A}\mathbf{x}_t + \mathbf{A}\mathbf{B}\mathbf{u}_t \mathrm{d}t = \overline{\mathbf{A}}\mathbf{x}_t + \overline{\mathbf{A}\mathbf{B}}\mathbf{u}_t \quad \text{assuming} \quad \mathrm{d}\mathbf{u}_t = \mathbf{0} \in [t, t'], \tag{15}$$

where the resulting discretized matrices are

$$\overline{\mathbf{A}} = e^{\mathbf{A}\Delta t}, \quad \overline{\mathbf{A}\mathbf{B}} = \mathbf{A}^{-1}(e^{\mathbf{A}\Delta t} - \mathbf{I})\mathbf{A}\mathbf{B}, \quad \text{where} \quad \Delta t = t' - t. \tag{16}$$

The ZOH does not affect the output or passthrough matrices $\mathbf{C}$ and $\mathbf{D}$. To compute the matrices $\overline{\mathbf{A}}$ and $\overline{\mathbf{A}\mathbf{B}}$ however requires computing a matrix exponential and a matrix inverse. Smith et al. [2022] avoid this by diagonalizing the system (also avoiding a dense matrix-matrix multiplication in the parallel scan). The diagonalized dynamics and input matrices are denoted $\mathbf{\Lambda}$ (a diagonal matrix) and $\mathbf{\Lambda}\tilde{\mathbf{B}}$ respectively. In this case, Eq. (16) reduces to

$$\overline{\mathbf{A}} = e^{\mathbf{\Lambda}\Delta t}, \tag{17}$$

$$\overline{\mathbf{A}\mathbf{B}} = \mathbf{\Lambda}^{-1}(e^{\mathbf{\Lambda}\Delta t} - \mathbf{I})\mathbf{\Lambda}\tilde{\mathbf{B}} \tag{18}$$

$$= (e^{\mathbf{\Lambda}\Delta t} - \mathbf{I})\tilde{\mathbf{B}} \quad \text{(diagonal matrices commute)} \tag{19}$$

where $e^{\mathbf{\Lambda}\Delta t}$ is trivially computable as the exponential of the leading diagonal of $\mathbf{\Lambda}\Delta t$. These operations are embarrassingly parallelizable across the sequence length and state dimension given the desired evaluation times.

To contextualize, suppose an event occurs at time $t$, Eq. (19) allows us to exactly (under the constant-input assumption) efficiently evaluate the linear recurrence at subsequent times $t'$. We use this extensively in S2P2 to efficiently evaluate the recurrence (and hence the intensity) at the irregularly-spaced event times and times used to compute the integral term.

It should be noted the discretization was done to compute a left-limit $\mathbf{x}_{t'-}$ from a previous right-limit $\mathbf{x}_t$. Should an event not occur at $t'$, then the left- and right-limits agree and $\mathbf{x}_{t'-} = \mathbf{x}_{t'+} = \mathbf{x}_{t'}$. If an event does occur at time $t'$ with mark $k$, then the left-limit $\mathbf{x}_{t'-}$ can be incremented by $\tilde{\mathbf{E}}\boldsymbol{\alpha}_k$ to compute $\mathbf{x}_{t'+} = \mathbf{x}_{t'}$. This increment is exact and leverages no discretization assumption.

## B.3 Interpretation for Input-Dependent Dynamics

Consider the input-dependent recurrence for an LLH layer, as defined in Eq. (13):

$$\mathrm{d}\tilde{\mathbf{x}}_t := \mathbf{\Lambda}_i \tilde{\mathbf{x}}_{t-} \mathrm{d}t + \mathbf{\Lambda}_i \tilde{\mathbf{B}}\mathbf{u}_{t-} \mathrm{d}t + \tilde{\mathbf{E}}\boldsymbol{\alpha}\mathrm{d}\mathbf{N}_t \tag{20}$$

for $t \in (t_i, t_{i+1}]$ where $\mathbf{\Lambda}_i := \mathrm{diag}(\Delta_i)\mathbf{\Lambda}$ with the input-dependent factor defined as $\Delta_i := \mathrm{softplus}(\mathbf{W}'\mathbf{u}_{t_i} + \mathbf{b}') \in \mathbb{R}_{>0}^P$. This factor can be thought of as the input-dependent relative-time scale for the dynamics. To see this, we first note that for vectors $\mathbf{p}, \mathbf{q} \in \mathbb{R}^d$, the following holds true: $\mathrm{diag}(\mathbf{p})\mathbf{q} = \mathbf{p} \odot \mathbf{q} = \mathbf{q} \odot \mathbf{p}$ where $\odot$ is the Hadamard or element-wise product. It then follows that

$$\mathrm{d}\tilde{\mathbf{x}}_t := \mathbf{\Lambda}_i \tilde{\mathbf{x}}_{t-} \mathrm{d}t + \mathbf{\Lambda}_i \tilde{\mathbf{B}}\mathbf{u}_{t-} \mathrm{d}t + \tilde{\mathbf{E}}\boldsymbol{\alpha}\mathrm{d}\mathbf{N}_t \tag{21}$$

$$= \mathbf{\Lambda}_i (\tilde{\mathbf{x}}_{t-} + \tilde{\mathbf{B}}\mathbf{u}_{t-})\mathrm{d}t + \tilde{\mathbf{E}}\boldsymbol{\alpha}\mathrm{d}\mathbf{N}_t \tag{22}$$

$$= \mathrm{diag}(\Delta_i)\mathbf{\Lambda}(\tilde{\mathbf{x}}_{t-} + \tilde{\mathbf{B}}\mathbf{u}_{t-})\mathrm{d}t + \tilde{\mathbf{E}}\boldsymbol{\alpha}\mathrm{d}\mathbf{N}_t \tag{23}$$

$$= [\mathbf{\Lambda}(\tilde{\mathbf{x}}_{t-} + \tilde{\mathbf{B}}\mathbf{u}_{t-})] \odot (\Delta_i \mathrm{d}t) + \tilde{\mathbf{E}}\boldsymbol{\alpha}\mathrm{d}\mathbf{N}_t. \tag{24}$$

As shown, the positive vector $\Delta_i$ can be thought of as changing the relative time-scale for each channel in the hidden state $\tilde{\mathbf{x}}$. Large values of $\Delta_i$ will act as if time is passing quickly, encouraging the state to converge to the steady-state sooner. Conversely, smaller values will make time pass more slowly causing the model to retain the influence that prior events have on future ones (for that specific channel in $\tilde{\mathbf{x}}$ at least).

## B.4 Forwards and Backwards Zero Order Hold Discretization

In Section 3.2 we highlighted that the ZOH discretization is exact when $\mathbf{u}_t$ is held constant over the integration window. This raises a unique design question for S2P2: what constant value should $\mathbf{u}_t$ take on when evolving $\mathbf{x}$ from time $t$ to $t'$? For the first layer of the model, the input is zero by construction, so there is no choice to be made—in fact, since $\mathbf{u}$ is constant for the first layer the updates are exact. However, the input is non-zero at deeper layers, and, crucially, varies over the integration period.

We must therefore decide how to select a $\mathbf{u}$ value over the integration period. This should be a value in (or function of) $\{\mathbf{u}_s \mid s \in [t, t')\}$. Note this is because the value at $t'$, $\mathbf{u}_{t'}$, *cannot* be incorporated as this would cause a data leakage in our model; while values prior to $t$ would discard the most recent event occurrence. For this work, we explore two natural choices: (i) the input value at the beginning of the interval, $\mathbf{u}_t$, and (ii) the left-limit at the end of the interval, $\mathbf{u}_{t'-}$. Note the end of the interval need not align with an event (crucial for when computing intermediate intensity values). We refer to these options as *forwards* and *backwards* ZOH, respectively. We illustrate the backwards variant in Fig. 2, where in the leftmost panel, we use the $\mathbf{u}_{t-}$ values at each layer to calculate $\mathbf{x}_{t-}$ and $\mathbf{x}_t$, as opposed to $\mathbf{u}_{t_i}$ for $t_i < t < t_{i+1}$. All experiments in the main paper utilize backwards ZOH.

It is not obvious *a priori* which one of these modes is more performant. We therefore conducted an ablation experiment in Table 14. We see that there is little difference between the two methods. We also note that models are learned *through* this discretization, and so this decision does not mean that a model is "incorrectly discretized" one way or the other, but instead they define subtlety different families of models. Theoretical and empirical investigation of the interpretations of this choice is an interesting area of investigation going forwards, extending the ablations we present in Table 14.

## B.5 Theoretical Complexity

We include in Table 5 a brief summary of the theoretical complexity of each of the methods we consider, broadly grouped by their architectures. We analyze complexity by the work, memory complexity and theoretical best parallel application time of the forward pass (used when conditioning on a sequence, the left-hand term of Eq. (2)) and evaluating the integral term in Eq. (2) *given that the forward pass has been completed* (as this is either required by the method, and is nearly always evaluated in conjunction with the forward pass). We then state the limiting best-case theoretical parallelism of the two components.

The reasoning behind the calculated values are as follows:

- The forward pass of RMTPP, NHP and IFTPP use non-linear RNNs, and hence incur memory and work that is linear in the sequence length, and cannot be parallelized. MHP uses an RNN, but that is logarithmically parallelizable. These models re-use the computed hidden states to compute the integral term, and hence, while they incur work and memory that scales in the sequence length and number of events, this work can be perfectly parallelized. This results in a best-case parallelism of $\mathcal{O}(N)$ (dominated by the forward pass; $\mathcal{O}(\log N)$ for MHP).

- SAHP, THP and AttNHP all use self-attention, and hence have a work and memory that scales quadratically in the sequence length, although this work can be parallelized across the sequence length, resulting in logarithmic parallel depth. SAHP and THP re-use embeddings and a parametric decoder, and hence estimating the integral scales like the RNN, and hence the limiting parallelism is still the forward pass. AttNHP is slightly different in that it re-applies the whole independently attention mechanism for each integration point. However, this work is parallelizeable and hence still reduces to a best-case depth of $\mathcal{O}(\log N)$.

- S2P2 is an RNN and hence has linear work and memory in the forward pass, but can be parallelized to a best-case depth of $\mathcal{O}(\log N)$ using the parallel scan. We then re-use the states computed in the forward pass for estimating the integral, which, as with the other RNN methods, is perfectly parallelizable, resulting in a theoretical parallel depth of $\mathcal{O}(\log N)$.

Note that these figures do not account for the number of layers required by each model, which must be evaluated in sequence.

Table 5: Comparison of methods based on memory and compute complexity. We see that our S2P2 matches the best performing baseline in all categories. $N$ denotes to the sequence length, and $M$ denotes to the number of Monte Carlo grid points per-event used in evaluating Eq. (2). As IFTPP is an intensity-free method, it does not need to estimate $\int \lambda_t \mathrm{d}t$ as the other methods do.

| Method | **Forward Pass** | | | **Estimating $\int \lambda_t \mathrm{d}t$** | | | **Overall** |
| | Memory | Work | Theoretical Parallelism | Memory | Work | Theoretical Parallelism | Theoretical Parallelism |
|---|---|---|---|---|---|---|---|
| RMTPP | $\mathcal{O}(N)$ | $\mathcal{O}(N)$ | $\mathcal{O}(N)$ | $\mathcal{O}(NM)$ | $\mathcal{O}(NM)$ | $\mathcal{O}(1)$ | $\mathcal{O}(N)$ |
| NHP | $\mathcal{O}(N)$ | $\mathcal{O}(N)$ | $\mathcal{O}(N)$ | $\mathcal{O}(NM)$ | $\mathcal{O}(NM)$ | $\mathcal{O}(1)$ | $\mathcal{O}(N)$ |
| MHP | $\mathcal{O}(N)$ | $\mathcal{O}(N)$ | $\mathcal{O}(\log N)$ | $\mathcal{O}(NM)$ | $\mathcal{O}(NM)$ | $\mathcal{O}(1)$ | $\mathcal{O}(\log N)$ |
| IFTPP | $\mathcal{O}(N)$ | $\mathcal{O}(N)$ | $\mathcal{O}(N)$ | N/A | N/A | N/A | $\mathcal{O}(N)$ |
| SAHP | $\mathcal{O}(N^2)$ | $\mathcal{O}(N^2)$ | $\mathcal{O}(\log N)$ | $\mathcal{O}(NM)$ | $\mathcal{O}(NM)$ | $\mathcal{O}(1)$ | $\mathcal{O}(\log N)$ |
| THP | $\mathcal{O}(N^2)$ | $\mathcal{O}(N^2)$ | $\mathcal{O}(\log N)$ | $\mathcal{O}(NM)$ | $\mathcal{O}(NM)$ | $\mathcal{O}(1)$ | $\mathcal{O}(\log N)$ |
| AttNHP | $\mathcal{O}(N^2)$ | $\mathcal{O}(N^2)$ | $\mathcal{O}(\log N)$ | $\mathcal{O}(N^2M)$ | $\mathcal{O}(N^2M)$ | $\mathcal{O}(\log N)$ | $\mathcal{O}(\log N)$ |
| S2P2 | $\mathcal{O}(N)$ | $\mathcal{O}(N)$ | $\mathcal{O}(\log N)$ | $\mathcal{O}(NM)$ | $\mathcal{O}(NM)$ | $\mathcal{O}(1)$ | $\mathcal{O}(\log N)$ |

To validate the scaling properties, we measure the wallclock time for a full forward pass and log-likelihood evaluation on random input sequences with lengths ranging from eight events to over half a million events. The architectures and mark spaces are the same as in the StackOverflow experiments (see Tables 6a and 7).

We note: our `EasyTPP` PyTorch S2P2 is written in pure PyTorch, and hence is not as optimized as other methods (compared to, for instance, IFTPP, which uses a GPU-optimized implementation of the GRU). We therefore include the runtimes of a standalone JAX S2P2 implementation, which allows for comparable levels of optimization through JIT compilation.

We observe the predicted scaling in practice. NHP scales linearly across all sequence lengths, and is far outpaced by all other methods. The THP scales well before reverting to superlinear scaling, and then runs out of memory. The IFTPP is very fast at shorter runtimes, but quickly reverts to linear scaling, due to its simple but highly optimized implementation and inherently sequential operation. Both S2P2 implementations scale linearly at long sequence lengths, but have near-constant runtime at shorter sequences. At shorter sequence lengths, the more optimized JAX implementation is faster than the unoptimized pure PyTorch implementation. While this indicates that there are additional opportunities to accelerate the PyTorch implementation further (e.g., exploiting kernel fusion or writing a lower-level Triton implementation), these results still confirm that our S2P2 can exploit parallel scans to scale to long sequences more effectively than other methods while retaining strong performance.

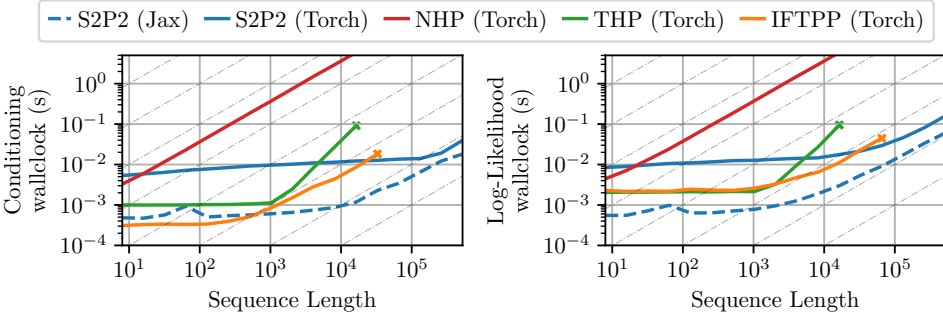

Figure 5: Median runtime of various models against increasing sequence lengths when conditioning on a sequence (Algorithm 1) and for likelihood evaluation (Algorithm 3) over 20 random seeds (variance negligible). S2P2 is faster across a wide range of sequence lengths. Crosses indicate where THP runs out of memory or IFTPP throws an error.

# C Experimental Configurations and Datasets

## C.1 Training Details & Hyperparameter Configurations

All baseline models used up-to-date PyTorch implementations, provided by the `EasyTPP` library [Xue et al., 2023] as of May 2025.

We apply a grid search for all models on all datasets for hyperparameter tuning. We use a default batch size of 256 for training. For models/datasets that require more memory (e.g., large mark space or long sequences), we reduce the batch size and keep them as consistent as possible among all the models on each dataset. We use the Adam stochastic gradient optimizer [Kingma and Ba, 2015], with a learning rate of 0.01 and a linear warm-up schedule over the first 1% iterations, followed by a cosine decay. Initial experiments showed this setting generally worked well across different models and datasets leads to convergence within 300 epochs. We also clip the gradient norm to have a max norm of 1 for training stability. We use Monte-Carlo samples to estimate the integral in log-likelihood, where we use 10 Monte-Carlo points per event during training.

On the five `EasyTPP` benchmark datasets and MIMIC-II that are smaller in their scales, we choose an extended grid based on the architecture reported in the `EasyTPP` paper. Specifically, we search over hidden states size $h = \{16, 32, 64, 128, 256\}$ for RMTPP, $h = \{32, 64, 128\}$ for NHP, and $h = \{16, 32, 64\}$ for IFTPP. For SAHP, THP, and AttNHP, we searched over all combinations of number of $L = \{1, 2, 3\}$, hidden state size $= \{16, 32, 64, 128\}$, and number of heads $= \{1, 2, 4\}$. For MHP, we followed their paper to fix $L = 4$, then we search over $h = \{4, 8, 16, 32, 64, 128, 256, 512\}$. Finally, for S2P2, we considered combinations for number of layers $= \{1, 2, 3, 4\}$, $p = \{16, 32, 64, 128\}$ and $h = \{16, 32, 64, 256\}$. We generally found a range of reasonable hyperparameters to yield similar performance on S2P2, while multiple layers were key for performance (intuitively deep stack allows nonlinear and complex dynamics), but there was not a critical dependence on depth.

We fixed the activation function as GeLU [Hendrycks and Gimpel, 2016] and apply post norm with layer norm [Ba, 2016]. We fix the dropout as 0.1 for S2P2 on the five core benchmark datasets, and add dropout $= \{0, 0.1\}$ to the grid search for the other three datasets. Due to the scale of Last.fm and EHRSHOT datasets, we perform a smaller search over architectures that roughly match the parameter counts for all models at three levels: 25k, 50k, 200k, and choose the model with the best validation results. AttNHP has expensive memory requirements that tends to have smaller batch sizes than other models. We were unable to train any AttNHP on EHRSHOT. The final model architectures used are reported in Tables 6a and 6b.

Table 6: Model architectures for the five `EasyTPP` benchmark datasets in Table 2.

(a) Model architectures for the five `EasyTPP` benchmark datasets in Table 2.

| Model | Amazon | Retweet | Taxi | Taobao | StackOverflow |
|---|---|---|---|---|---|
| RMTPP | $h = 128$ | $h = 16$ | $h = 128$ | $h = 16$ | $h = 256$ |
| SAHP | $h = 32, l = 2, \text{heads} = 2$ | $h = 32, l = 3, \text{heads} = 4$ | $h = 16, l = 2, \text{heads} = 4$ | $h = 32, l = 1, \text{heads} = 1$ | $h = 64, l = 1, \text{heads} = 1$ |
| THP | $h = 32, l = 2, \text{heads} = 4$ | $h = 16, l = 3, \text{heads} = 4$ | $h = 128, l = 1, \text{heads} = 4$ | $h = 64, l = 1, \text{heads} = 1$ | $h = 16, l = 2, \text{heads} = 4$ |
| IFTPP | $h = 64$ | $h = 64$ | $h = 32$ | $h = 64$ | $h = 64$ |
| MHP | $h = 8$ | $h = 16$ | $h = 4$ | $h = 16$ | $h = 8$ |
| NHP | $h = 128$ | $h = 64$ | $h = 128$ | $h = 128$ | $h = 64$ |
| AttNHP | $h = 64, t = 16, l = 2, \text{heads} = 4$ | $h = 16, t = 16, l = 2, \text{heads} = 4$ | $h = 16, t = 16, l = 3, \text{heads} = 4$ | $h = 32, t = 16, l = 3, \text{heads} = 4$ | $h = 32, t = 16, l = 2, \text{heads} = 4$ |
| S2P2 | $h = 64, p = 128, l = 2$ | $h = 128, p = 128, l = 2$ | $h = 128, p = 16, l = 4$ | $h = 32, p = 16, l = 4$ | $h = 32, p = 32, l = 3$ |

(b) Model architectures for the additional three benchmark datasets in Table 2.

| Model | Last.fm | MIMIC-II | EHRSHOT |
|---|---|---|---|
| RMTPP | $h = 256$ | $h = 128$ | $h = 16$ |
| SAHP | $h = 136, l = 2, \text{heads} = 4$ | $h = 64, l = 2, \text{heads} = 4$ | $h = 8, l = 2, \text{heads} = 4$ |
| THP | $h = 48, l = 2, \text{heads} = 4$ | $h = 32, l = 3, \text{heads} = 4$ | $h = 32, l = 2, \text{heads} = 4$ |
| IFTPP | $h = 48$ | $h = 256$ | $h = 16$ |
| MHP | $h = 16$ | $h = 16$ | $h = 32$ |
| NHP | $h = 112$ | $h = 128$ | $h = 80$ |
| AttNHP | $h = 28, t = 16, l = 2, \text{heads} = 4$ | $h = 64, t = 16, l = 3, \text{heads} = 2$ | OOM |
| S2P2 | $h = 68, p = 16, l = 2$ | $h = 64, p = 16, l = 2$ | $h = 128, p = 32, l = 2$ |

## C.2 Dataset Statistics

We report the statistics of all eight datasets we used in Table 7. We used the `HuggingFace` version of the five `EasyTPP` datasets. For all datasets, we further ensure the MTPP modeling assumptions are satisfied that no more than two events occur at the same time (i.e., inter-arrival time is strictly positive), and event times do not lie on grid points that are effectively discrete-time events. Dataset descriptions and pre-processing details are provided in Appendix C.3.

Table 7: Statistics of the eight datasets we experiment with.

| Dataset | $K$ | Number of Events | | | Sequence Length | | | Number of Sequences | | |
|---|---|---|---|---|---|---|---|---|---|---|
| | | Train | Valid | Test | Min | Max | Mean | Train | Valid | Test |
| Amazon | 16 | 288,377 | 40,995 | 84,048 | 14 | 94 | 44.8 | 6,454 | 922 | 1,851 |
| Retweet | 3 | 2,176,116 | 215,521 | 218,465 | 50 | 264 | 108.8 | 20,000 | 2,000 | 2,000 |
| Taxi | 10 | 51,584 | 7,404 | 14,820 | 36 | 38 | 37.0 | 1,400 | 200 | 400 |
| Taobao | 17 | 73,483 | 11,472 | 28,455 | 28 | 64 | 56.7 | 1,300 | 200 | 500 |
| StackOverflow | 22 | 90,497 | 25,762 | 26,518 | 41 | 101 | 64.8 | 1,401 | 401 | 401 |
| Last.fm | 120 | 1,534,738 | 344,542 | 336,676 | 6 | 501 | 207.2 | 7,488 | 1,604 | 1,604 |
| MIMIC-II | 75 | 9,619 | 1,253 | 1,223 | 2 | 33 | 3.7 | 2600 | 325 | 325 |
| EHRSHOT | 668 | 759,141 | 165,237 | 170,147 | 5 | 3,955 | 177.0 | 4,329 | 927 | 927 |

## C.3 Dataset Pre-processing

We use the default train/validation/test splits for `EasyTPP` benchmark datasets. For MIMIC-II, we copy Du et al. [2016] and keep the 325 test sequences in the test split, and further split the 2,935 training sequences into 2,600 for training and 325 for validation. In our pre-processed datasets, Last.fm and EHRSHOT, we randomly partition into subsets containing 70%, 15%, 15% of all sequences for training/validation/test respectively. We provide a high-level description of all the datasets we used, followed by our pre-processing procedure of Last.fm and EHRSHOT in more detail. Note that for datasets that contain concurrent events or effectively discrete times (e.g., StackOverflow, Retweet), we apply a small amount of jittering to ensure no modeling assumptions are violated in the MTPP framework.

**Amazon** [Ni et al., 2019] contains user product reviews where product categories are considered as marks. **Retweet** [Zhao et al., 2015] predicts the popularity of a retweet cascade, where the event type is decided by if the retweet comes from users with "small", "medium", or "large" influences, measured by number of followers [Mei and Eisner, 2017]. **Taxi** data [Whong, 2014, Mei et al., 2019] uses data from the pickups and dropoffs of New York taxi and the marks are defined as the Cartesian product of five discrete locations and two actions (pickup/dropoff). **Taobao** [Xue et al., 2022] describes the viewing patterns of users on an e-commerce site, where item categories are considered as marks. **StackOverflow** contains the badges (defined as marks) awarded to users on a question-answering website. Finally, **MIMIC-II** [Saeed et al., 2002] records different diseases (used as marks) during hospital visits of patients. We add a small amount of noise to the MIMIC-II event times so that events do not lie on a fixed grid. Both StackOverflow and MIMIC-II datasets were first pre-processed by Du et al. [2016].

**Last.fm** [Celma, 2010, McFee et al., 2012] records 992 users' music listening habits that has been widely used in MTPP literature [Kumar et al., 2019, Boyd et al., 2020, Bosser and Taieb, 2023]. Mark types are defined as the genres of a song, and each event is a play of a particular genre. Each sequence represents the monthly listening behavior of each user, with sequence lengths between 5 and 500. If the song is associated with multiple genres we select a random one of the genres, resulting in a total of 120 different marks.

**EHRSHOT** [Wornow et al., 2023] is a newly proposed large dataset of longitudinal de-identified patient medical records, and has rich information such as hospital visits, procedures, and measurements. We introduce an MTPP dataset derived from EHRSHOT, where medical services and procedures are treated as marks, as identified by *Current Procedural Terminology* (CPT-4) codes. Each patient defines an event sequence, and we retain only CPT-4 codes with at least 100 occurrences in the dataset. For the $< 1\%$ events of events where there are more than 10 codes at a single timestamp, we retain the top 10 codes with the most frequencies and discard the rest. We then add a small amount of random noise to the event time to ensure they are not overlapping. This process ensures we still satisfy the MTPP framework, and can reasonably instead compute top-10 accuracy for the next mark prediction; other work has considered extending the MTPP framework to consider simultaneous event

occurrence [Chang et al., 2024]. Then we standardize each sequence to start at $t = 0$ and pad the start and end of sequences with a specific padded event token. Note that we do not score these events. Event times are normalized to be in hours. We discard sequences that have less than 5 events or a single timestamp. This leads to the final version of our dataset having 668 marks, and the sequence lengths range from 5 to 3955 events, reflecting patient histories that can span multiple years.

We include code in our forked repository for preparing the EHRSHOT event sequence dataset from the raw EHRSHOT dataset. Note that we cannot distribute the raw data (or derivative dataset) under the terms of the original EHRSHOT dataset requiring credentialed access through PhysioNet.

# D Additional Experimental Results

## D.1 Full Results on Benchmark Datasets

We provide the full log-likelihood results and corresponding plots in Table 8 and Fig. 6 respectively, where we decompose the likelihood into time and mark likelihoods. The improvement of our S2P2 model is mainly driven by better modeling of time, though we also often obtain best- or second-best predictive performance on marks from the next event prediction accuracy results conditioned on true event time in Table 2c. In aggregate, our model achieves a 1.33 per-event likelihood ratio between itself and the next best method across all datasets (a 33% improvement in likelihood). This is calculated by computing the mean log-likelihood ratio across all datasets and then exponentiating. Doing so is equivalent to taking the geometric mean across likelihood ratios.

Configuration and training details of all models can be found in Appendix C.1. As discussed in Section 7 and grouped in the results, models with continuous-time hidden states can present a richer class of intensities and often empirically outperform those with discrete hidden states. Note for the RMSE results in Table 2b, we follow Mei and Eisner [2017] and use the expected next event time as next event time predictions to minimize the Bayes risk. Unlike them, however, we estimate these with the trapezoidal rule rather than Monte-Carlo simulation via the thinning algorithm. In practice, we have found this to produce an estimator with much lower variance and be faster due to being more readily parallelizable. This stands in contrast to the thinning algorithm which has more hyperparameters (e.g., dominating rate, sampling boundary) that can exacerbate bias.

Table 8: Complete per event log-likelihood results on the held-out test for the eight benchmark datasets we consider, averaged over 5 random seeds. In Table 8a we show the full log-likelihood. We then decompose this log-likelihood into the log-likelihood of the event time in Table 8b, and the time-conditional log-likelihood of the mark type in Table 8c. OOM indicates out of memory; standard deviation in parentheses. We $\boxed{\text{box}}$ the best-performing model and underline the second-best. We also report the average rank of models across datasets as a summary metric (lower is better). S2P2 is consistently the best or second best-performing model across all datasets.

(a) Full log-likelihood results (equal to the summation of Table 8b and Table 8c). Extended version of Table 2a.

| Model | Per Event Log-Likelihood, $\mathcal{L}_{\text{Total}}$ (nats) (↑) | | | | | | | | Avg. Ranking (↓) |
|---|---|---|---|---|---|---|---|---|---|
| | Amazon | Retweet | Taxi | Taobao | StackOverflow | Last.fm | MIMIC-II | EHRSHOT | |
| RMTPP | -2.136 (0.003) | -7.098 (0.217) | 0.346 (0.002) | 1.003 (0.004) | -2.480 (0.019) | -1.780 (0.005) | -0.472 (0.026) | -8.081 (0.025) | 7.1 |
| SAHP | -2.074 (0.029) | -6.708 (0.029) | 0.298 (0.057) | 1.168 (0.029) | -2.341 (0.058) | -1.646 (0.083) | -0.677 (0.072) | -6.804 (0.126) | 5.8 |
| THP | -2.096 (0.002) | -6.659 (0.007) | 0.372 (0.002) | 0.790 (0.002) | -2.338 (0.014) | -1.712 (0.011) | -0.577 (0.011) | -7.208 (0.096) | 6.1 |
| IFTPP | 0.496 (0.002) | -10.344 (0.016) | 0.453 (0.002) | $\boxed{1.318}$ (0.017) | -2.233 (0.009) | $\boxed{-0.492}$ (0.017) | 0.317 (0.052) | -6.596 (0.240) | 3.0 |
| MHP | -2.091 (0.002) | -6.564 (0.015) | 0.370 (0.008) | 0.636 (0.004) | -2.346 (0.012) | -1.676 (0.004) | -0.351 (0.012) | -7.206 (0.407) | 5.9 |
| NHP | 0.129 (0.012) | $\boxed{-6.348}$ (0.000) | 0.514 (0.004) | 1.157 (0.004) | -2.241 (0.002) | -0.574 (0.011) | 0.060 (0.017) | -3.966 (0.058) | 3.0 |
| AttNHP | 0.484 (0.077) | -6.499 (0.028) | 0.493 (0.009) | 1.259 (0.022) | -2.194 (0.016) | -0.592 (0.051) | -0.170 (0.077) | OOM | 3.1 |
| S2P2 (Ours) | $\boxed{0.781}$ (0.011) | -6.365 (0.003) | $\boxed{0.522}$ (0.004) | 1.304 (0.039) | $\boxed{-2.163}$ (0.009) | -0.557 (0.046) | $\boxed{0.919}$ (0.069) | $\boxed{-2.512}$ (0.369) | $\boxed{1.4}$ |

(b) Per event log-likelihood of the event times (higher is better).

| Model | Per Event Next Event Time Log-Likelihood, $\mathcal{L}_{\text{Time}}$ (nats) (↑) | | | | | | | | Avg. Ranking (↓) |
|---|---|---|---|---|---|---|---|---|---|
| | Amazon | Retweet | Taxi | Taobao | StackOverflow | Last.fm | MIMIC-II | EHRSHOT | |
| RMTPP | 0.011 (0.001) | -6.191 (0.083) | 0.622 (0.002) | 2.428 (0.004) | -0.797 (0.005) | 0.256 (0.007) | -0.188 (0.016) | -1.913 (0.025) | 6.1 |
| SAHP | 0.115 (0.049) | -5.872 (0.062) | 0.645 (0.044) | 2.604 (0.031) | -0.703 (0.031) | 0.489 (0.078) | -0.244 (0.040) | -1.801 (0.049) | 4.9 |
| THP | -0.068 (0.002) | -5.874 (0.007) | 0.621 (0.002) | 2.242 (0.002) | -0.772 (0.006) | 0.220 (0.010) | -0.271 (0.004) | -1.921 (0.027) | 7.0 |
| IFTPP | 2.483 (0.001) | -9.500 (0.011) | $\boxed{0.735}$ (0.002) | 2.708 (0.018) | -0.662 (0.007) | $\boxed{1.277}$ (0.016) | 0.555 (0.050) | -2.640 (0.115) | 3.1 |
| MHP | -0.064 (0.002) | -5.774 (0.016) | 0.620 (0.006) | 2.093 (0.004) | -0.761 (0.006) | 0.230 (0.003) | -0.140 (0.008) | -2.119 (0.318) | 6.4 |
| NHP | 2.116 (0.009) | $\boxed{-5.584}$ (0.001) | 0.727 (0.003) | 2.578 (0.006) | -0.699 (0.002) | 1.198 (0.006) | 0.225 (0.016) | -0.821 (0.045) | 3.3 |
| AttNHP | 2.416 (0.092) | -5.726 (0.027) | 0.714 (0.010) | 2.654 (0.007) | -0.684 (0.005) | 1.203 (0.015) | 0.031 (0.055) | OOM | 3.3 |
| S2P2 (Ours) | $\boxed{2.652}$ (0.009) | -5.598 (0.002) | 0.733 (0.003) | $\boxed{2.719}$ (0.038) | $\boxed{-0.641}$ (0.003) | 1.257 (0.022) | $\boxed{1.050}$ (0.065) | $\boxed{0.382}$ (0.362) | $\boxed{1.4}$ |

(c) Per event log-likelihood of mark type conditioned on the arrival time (higher is better).

| Model | Per Event Next Mark Log-Likelihood, $\mathcal{L}_{\text{Mark}}$ (nats) (↑) | | | | | | | | Avg. Ranking (↓) |
|---|---|---|---|---|---|---|---|---|---|
| | Amazon | Retweet | Taxi | Taobao | StackOverflow | Last.fm | MIMIC-II | EHRSHOT | |
| RMTPP | -2.147 (0.003) | -0.908 (0.141) | -0.276 (0.000) | -1.425 (0.002) | -1.683 (0.015) | -2.035 (0.004) | -0.284 (0.014) | -6.168 (0.025) | 6.8 |
| SAHP | -2.189 (0.030) | -0.836 (0.036) | -0.346 (0.024) | -1.436 (0.027) | -1.638 (0.032) | -2.136 (0.070) | -0.433 (0.031) | -5.003 (0.132) | 6.9 |
| THP | -2.028 (0.002) | -0.785 (0.001) | -0.249 (0.001) | -1.451 (0.000) | -1.566 (0.008) | -1.932 (0.006) | -0.306 (0.009) | -5.287 (0.107) | 5.5 |
| IFTPP | -1.988 (0.001) | -0.844 (0.007) | -0.282 (0.001) | $\boxed{-1.391}$ (0.005) | -1.571 (0.003) | $\boxed{-1.769}$ (0.004) | -0.239 (0.002) | -3.956 (0.192) | 4.1 |
| MHP | -2.027 (0.001) | -0.790 (0.003) | -0.251 (0.004) | -1.456 (0.005) | -1.586 (0.006) | -1.906 (0.005) | -0.210 (0.005) | -5.087 (0.296) | 5.4 |
| NHP | -1.987 (0.003) | $\boxed{-0.764}$ (0.000) | -0.213 (0.002) | -1.421 (0.004) | -1.542 (0.001) | -1.772 (0.006) | -0.165 (0.002) | -3.144 (0.016) | 2.4 |
| AttNHP | -1.933 (0.024) | -0.773 (0.003) | -0.221 (0.002) | -1.395 (0.016) | $\boxed{-1.510}$ (0.013) | -1.795 (0.037) | -0.201 (0.025) | OOM | 2.4 |
| S2P2 (Ours) | $\boxed{-1.871}$ (0.002) | -0.767 (0.000) | $\boxed{-0.211}$ (0.002) | -1.415 (0.005) | -1.521 (0.008) | -1.814 (0.025) | $\boxed{-0.131}$ (0.014) | $\boxed{-2.893}$ (0.009) | $\boxed{1.9}$ |

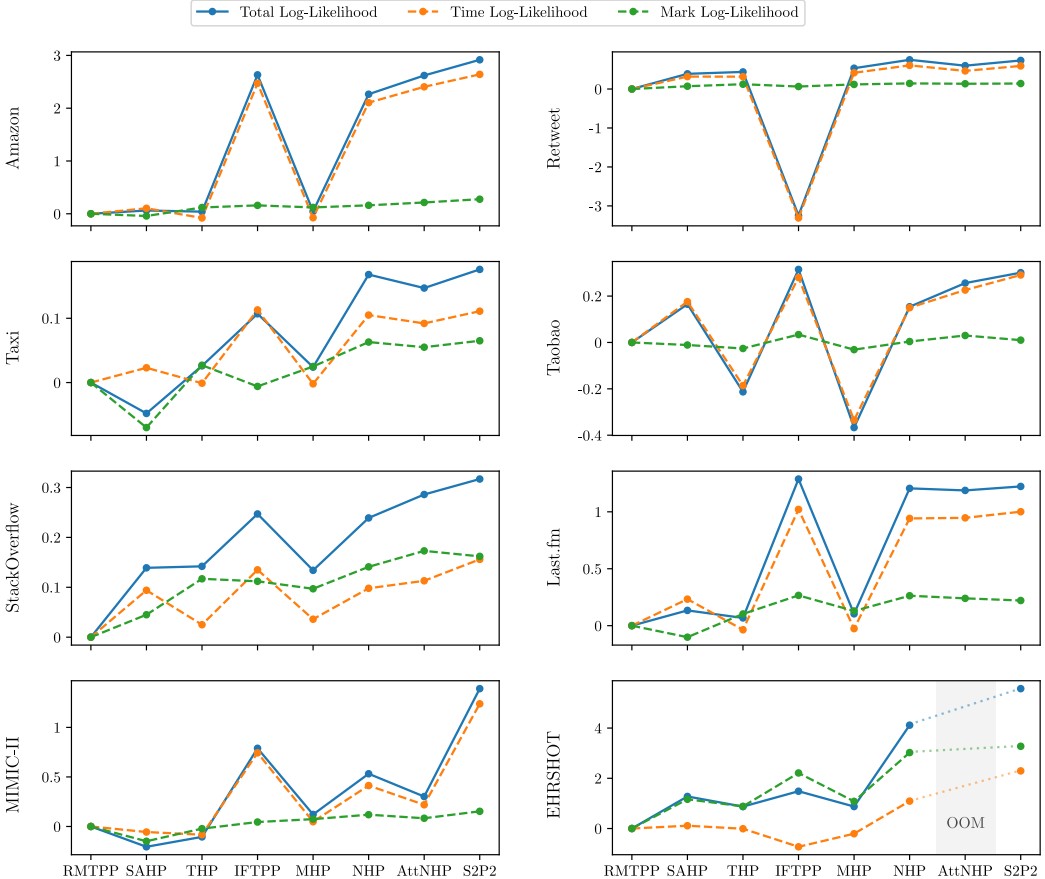

Figure 6: Visualization of $\mathcal{L}_{\text{Total}}$ decomposed into $\mathcal{L}_{\text{Time}}$ and $\mathcal{L}_{\text{Mark}}$ for all models and all datasets relative to RMTPP, normalized by number of events, as discussed in Section 6. The improvement of S2P2 is mainly driven by better modeling of $\mathcal{L}_{\text{Time}}$, while it improves both $\mathcal{L}_{\text{Time}}$ and $\mathcal{L}_{\text{Mark}}$.

### D.2 Full Results for Synthetic Poisson Experiments

We present the full results in Fig. 7 for all models regarding the synthetic Poisson experiments discussed in Section 5. All models are trained until convergence using a set of 5,000 generated sequences, where we use 20 Monte Carlo points per event to estimate the integral of log-likelihood during training to accommodate the sparsity of events. We used small models so they do not overfit; model architecture and parameter counts are reported in Table 9. We plot the estimated intensity conditioned on empty sequences using 1,000 equidistant grid points between the start and end points. Our model is the only one that perfectly recovers the underlying ground truth intensity, while also using the fewest parameters.

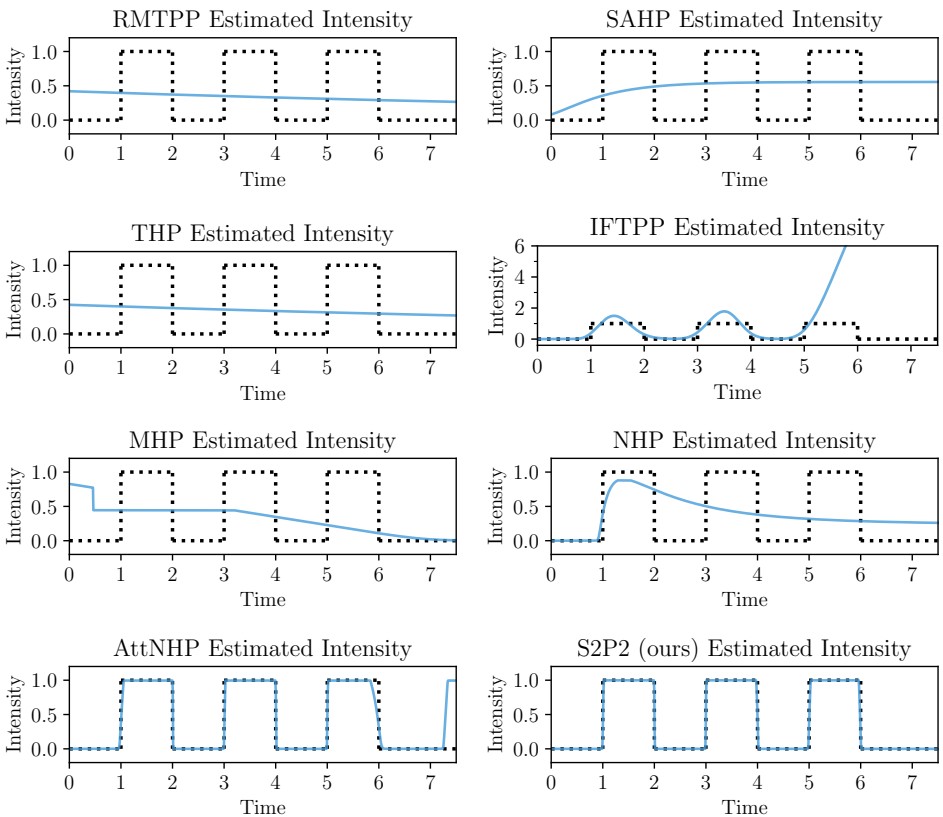

Figure 7: Results for all baseline models for the synthetic Poisson experiment introduced in Section 5. The estimated intensity (blue lines) conditioned on an empty sequence are plotted against the ground truth (dotted black lines).

Table 9: Model architectures and parameter counts for synthetic Poisson experiments.

| Model | Architecture | # Parameters |
|---|---|---|
| RMTPP | $h = 16$ | 627 |
| SAHP | $h = 16, l = 2, \text{heads} = 4$ | 1738 |
| THP | $h = 16, l = 2, \text{heads} = 4$ | 1684 |
| IFTPP | $h = 16$ | 1899 |
| MHP | $h = 4$ | 2240 |
| NHP | $h = 8$ | 1010 |
| AttNHP | $h = 8, t = 2, l = 2, \text{heads} = 2$ | 1178 |
| S2P2 (Ours) | $h = 4, p = 4, l = 2$ | 178 |

## D.3 Additional Synthetic Results on Multivariate Hawkes Processes

We evaluate our model and baseline models against the true model on a randomly initiated parametric Hawkes process with three possible marks. Following the notation in Section 2.1, we draw all parameters from the following distributions: $\nu_i \overset{iid}{\sim} \text{Unif}[0.1, 0.5]$, $\alpha_{ij} \overset{iid}{\sim} \text{Unif}[0.5, 0.8]$, and $\beta_{ij} \overset{iid}{\sim} \text{Unif}[0.4, 1.2]$ for $i, j \in \{1, 2, 3\}$.

All models are trained until convergence using a set of 50,000 generated sequences, where we use 20 Monte Carlo points per event to estimate the integral of log-likelihood during training. Model architecture and parameter counts are reported in Table 10. We plot three example sequences drawn for an additional test set for each model in Fig. 8, using 1,000 equidistant grid points for any inter-event interval. Dotted lines refer to the intensities under the true underlying parametric model; solid lines are different model estimates from trained models.

As we see in inhomogeneous Poisson processes, our model can recover the ground truth intensities with the fewest parameters. Visually, all SAHP, IFTPP, NHP, AttNHP and S2P2 (our model) perform well at recovering the ground truth intensities. It is also worth noting that our model is 7-9× quicker than NHP and AttNHP regarding wallclock runtime on a single A5000 GPU. Our results on synthetic experiments validate the model's ability to recover the ground truth intensities. We further evaluate all models quantitatively using 1,000 test sequences generated from the same multivariate Hawkes process and evaluated both log-likelihood and RMSE for the immediate next event. We see our method competitive again on both metrics.

Table 10: Model architectures and parameter counts for multivariate Hawkes processes experiments.

| Model | Architecture | # Parameters |
|-------|-------------|--------------|
| RMTPP | $h = 16$ | 697 |
| SAHP | $h = 16, l = 2, \text{heads} = 4$ | 1902 |
| THP | $h = 16, l = 2, \text{heads} = 4$ | 1756 |
| IFTPP | $h = 16$ | 1965 |
| MHP | $h = 4$ | 2264 |
| NHP | $h = 8$ | 1046 |
| AttNHP | $h = 8, t = 2, l = 2, \text{heads} = 2$ | 1230 |
| S2P2 (Ours) | $h = 8, p = 4, l = 2$ | 358 |

Table 11: Performance comparison of models on the multivariate Hawkes processes experiment presented above. Higher per-event log-likelihood indicates better performance, whereas lower root mean squared error (RMSE) indicates better performance.

| Model | Total Log-Likelihood $\mathcal{L}_{\text{Total}}$ ($\uparrow$) | Next-Event Time RMSE ($\downarrow$) |
|-------|-------------------------------------------------|-------------------------------------|
| RMTPP | -0.550 | 0.648 |
| SAHP | -0.537 | 0.647 |
| THP | -0.543 | 0.648 |
| IFTPP | -0.534 | 0.647 |
| MHP | -0.551 | 0.648 |
| NHP | -0.530 | 0.647 |
| AttNHP | -0.533 | 0.652 |
| S2P2 (Ours) | -0.527 | 0.647 |

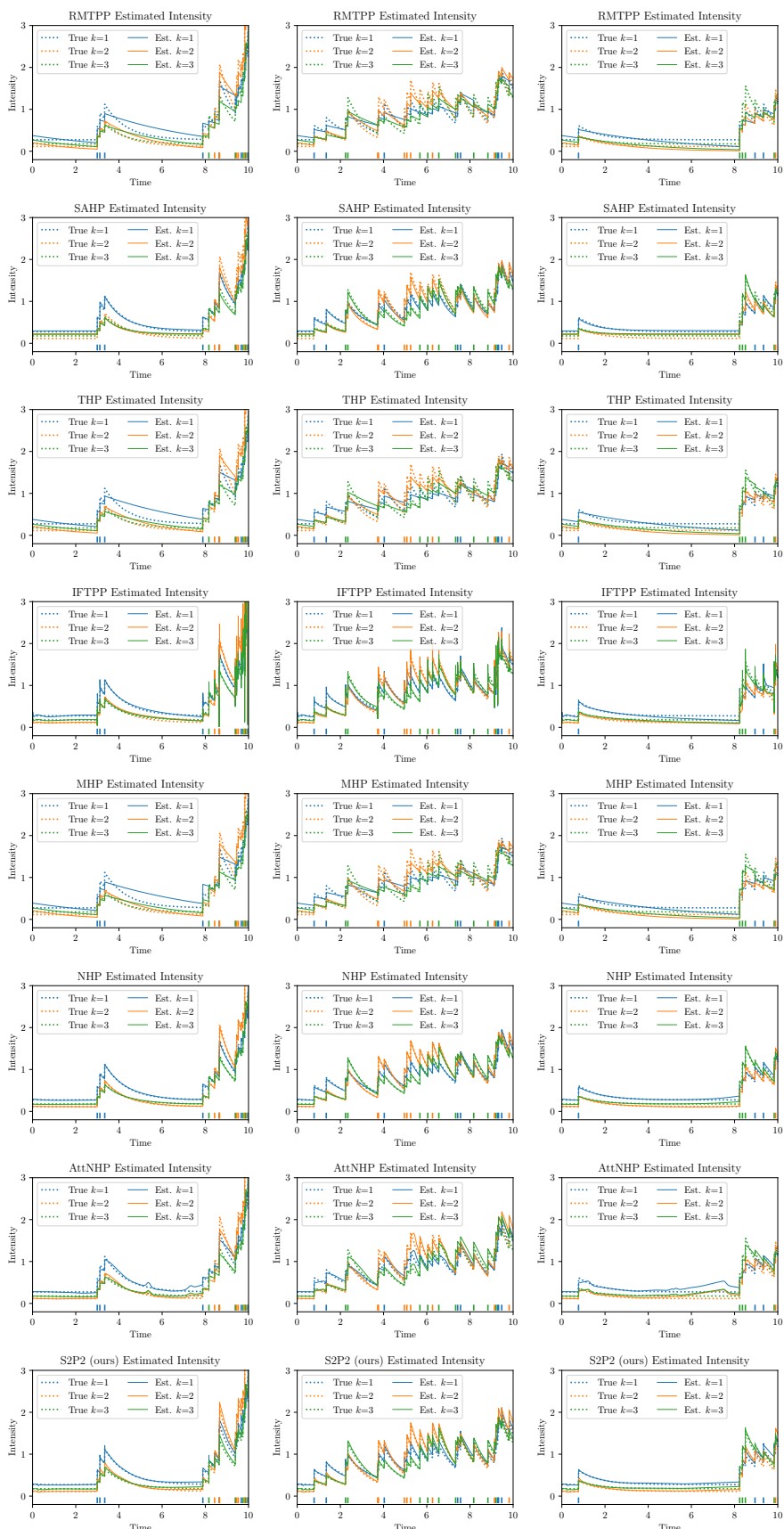

Figure 8: Our proposed S2P2 model and baseline models trained with 50k training sequences drawn from a randomly instantiated multivariate Hawkes process where $K = 3$. For each model, three example test sequences are plotted; the locations of colored bars ||| indicate the true event times.

### D.4   Full Results for Hawkes and Self-Correcting Process

We train our approach on synthetic data generated from known, classical temporal point processes, namely the self-correcting and Hawkes processes. These are characterized by intensity functions $\lambda_t^{\text{SC}} = \exp\left(t - 0.5N_t\right)$ and $\lambda_t^{\text{H}} = 0.5 + \sum_{i=1}^{N_t} 0.5 \exp(t_i - t)$, respectively. Models were fit using data drawn from these processes, 6,000 sequences for training and 2,000 for validation. The learned intensity functions, evaluated on a held-out test sequence, can be seen for the self-correcting process data in Fig. 9 and for the Hawkes process data in Fig. 10 for all methods. The hyperparameters for the models were chosen by a grid search (see Table 12). We see many of the models, including ours, do well at capturing the ground truth intensity.

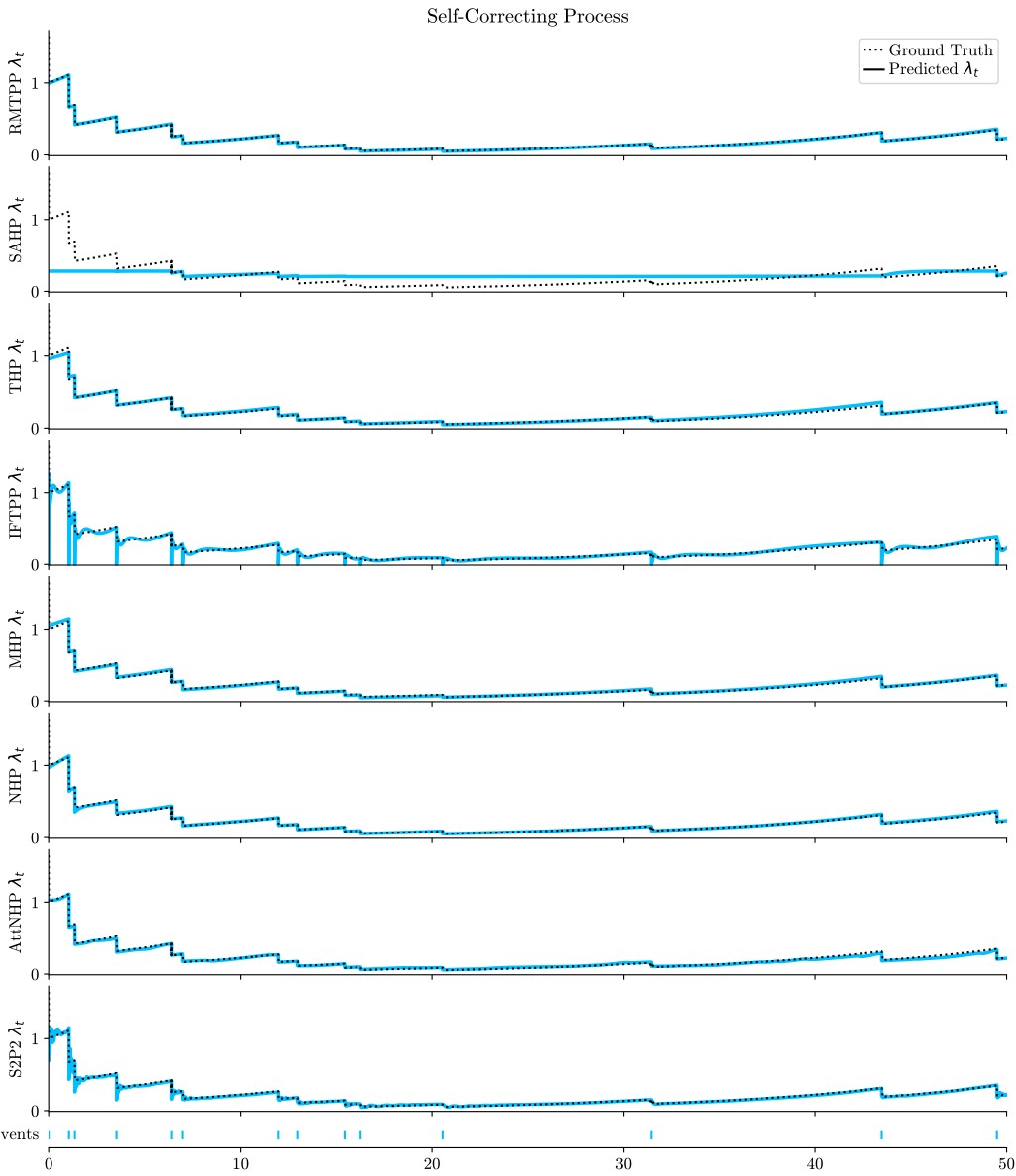

Figure 9: Synthetic self-correcting process experiment visualization of predicted intensities compared to the ground truth intensity for a given held-out sequence. The vertical lines present for IFTPP are due to the conversion from density to intensity being unstable near $\Delta t = 0$.

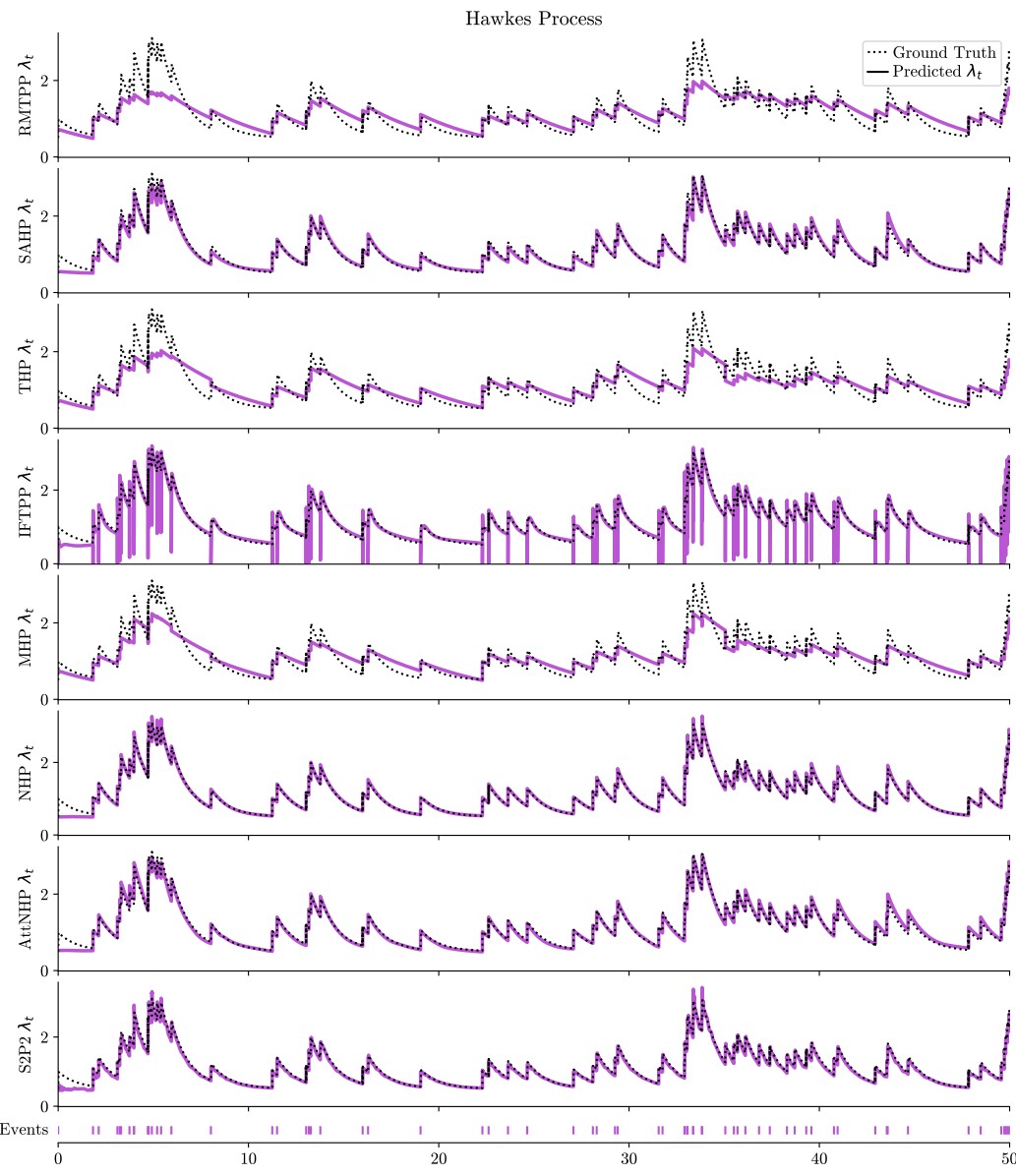

Figure 10: Synthetic Hawkes process experiment visualization of predicted intensities compared to the ground truth intensity for a given held-out sequence. The vertical lines present for IFTPP are due to the conversion from density to intensity being unstable near $\Delta t = 0$.

Table 12: Model architectures and parameter counts for synthetic Hawkes and self-correcting process experiments.

| Model | Architecture | # Parameters |
|---|---|---|
| RMTPP | $h = 16$ | 627 |
| SAHP | $h = 16, l = 3, \text{heads} = 4$ | 2554 |
| THP | $h = 16, l = 3, \text{heads} = 4$ | 2500 |
| IFTPP | $h = 32$ | 6859 |
| MHP | $h = 16$ | 13556 |
| NHP | $h = 32$ | 14658 |
| AttNHP | $h = 16, t = 16, l = 2, \text{heads} = 4$ | 13298 |
| S2P2 (Ours) | $h = 16, p = 16, l = 4$ | 7202 |

### D.5 Full Results for Long-Range Dependency Experiment

To measure the ability to capture long-range dependencies by neural MTPPs, we constructed a generative process with long-range dependencies. For this, we generate sequences over the time window of $[0, 100]$, with three possible marks. The first mark is a "distractor" mark, meaning it has no influence over other events. These events are drawn from a homogeneous Poisson process with rate 1. The second mark is a "trigger" mark, which are directly tied to the third "target" mark. Triggers are also drawn from a homogeneous Poisson process with rate 0.1. For every trigger event $(t_i, k_i = 2)$ drawn, a corresponding target event $(t_j, k_j = 3)$ is generated conditionally independent of all other events according to $t_j | t_i \sim \mathcal{N}(t_i + 40, 0.1)$.

All models were trained with the same hyperparameters as in Table 12. The predicted intensity functions for a single sequence can be seen in Fig. 11, and the likelihood ratio between the trained models and the ground truth process on held-out test sequences can be found in Table 13. We can see that both S2P2 and AttNHP do very well, both qualitatively and quantitatively. This is expected as both architectures are well suited for long-range dependencies while still being continuous-time models, allowing for expressive intensity functions.

Table 13: Likelihood ratios between models and ground truth process for held-out data on long-range experiment.

| Model | Ground Truth | RMTPP | SAHP | THP | IFTPP | MHP | NHP | AttNHP | S2P2 |
|---|---|---|---|---|---|---|---|---|---|
| Lik. Ratio | 100% | 80.4% | 81.0% | 94.0% | 88.0% | 87.3% | 87.9% | **99.7%** | 97.8% |

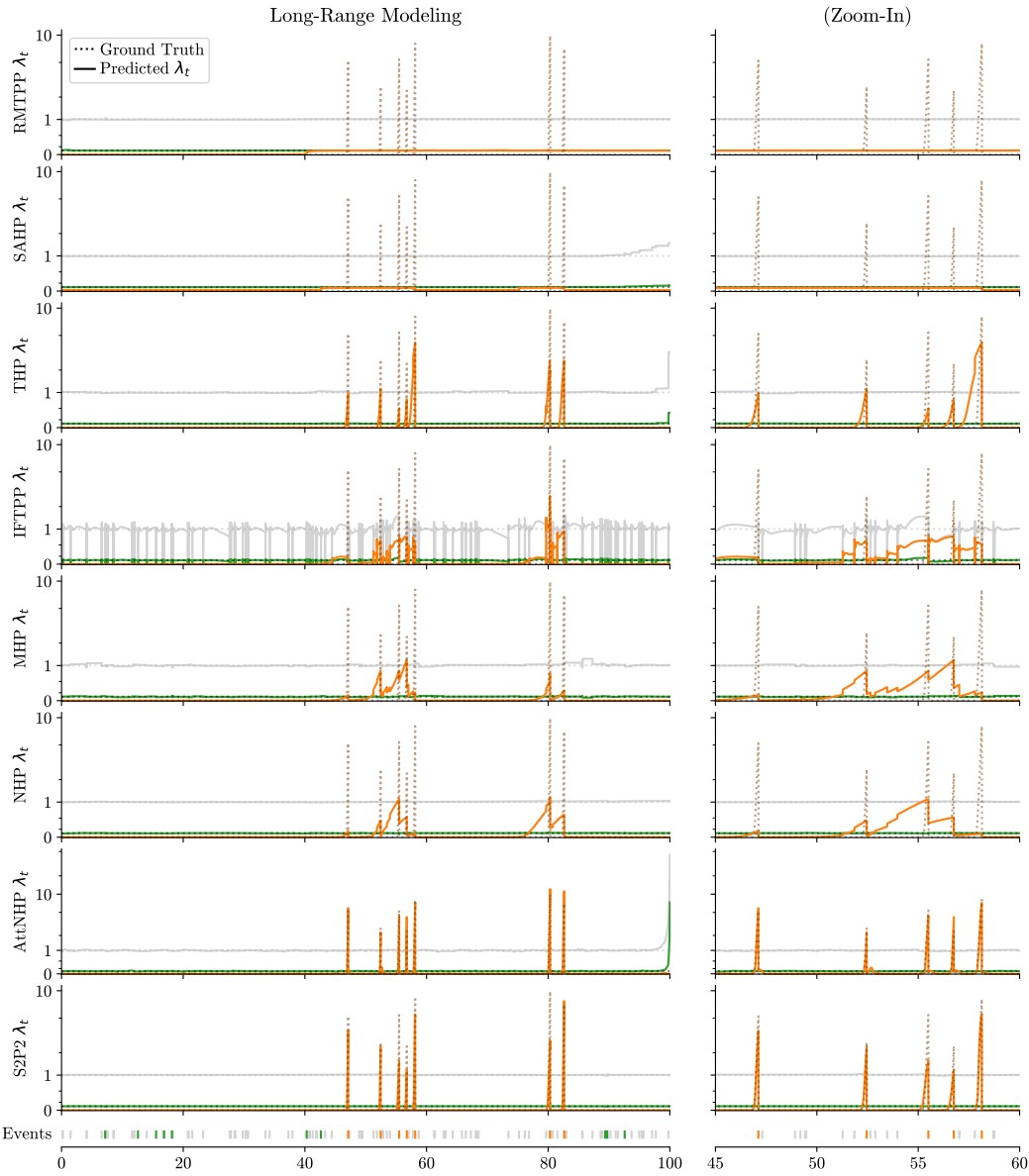

Figure 11: Synthetic long-range experiment visualization of predicted intensities compared to the ground truth intensity for a given held-out sequence. The vertical lines present for IFTPP are due to the conversion from density to intensity being unstable near $\Delta t = 0$.

## D.6 Ablation for Different S2P2 Variants

We perform an ablation study of different model variants that we proposed on all datasets and summarize the results in Table 14. We train EHRSHOT using 10% of its training data because larger dataset scale requires more training time (but use the original validation and test sets for model selection and reporting results). Forward and backward discretization are very close in performance, with backwards discretization having a slight edge. Models that are input-dependent achieve better performance on most datasets, although on certain datasets input dependence appears to harm performance. It is an interesting direction for future work to explore theoretically and empirically when each of these variants is best. We select backward discretization with input dependence for the results in the main paper.

Table 14: Ablation for different model variants log-likelihood (LL). ID stands for input-dependent, see Section 3.3. Backward and Forward respectively refer to using $\mathbf{u}_{t_{i-1}}$ and $\mathbf{u}_{t_i-}$ (i.e., the previous right limit or current left limit), see Appendix B.4.

| Dataset | Model variant | LL | Arrival time LL | Mark LL conditioned on time |
|---|---|---|---|---|
| Amazon | Forward | 0.705 | 2.617 | -1.912 |
| | Forward + ID | 0.748 | 2.634 | -1.886 |
| | Backward | 0.740 | 2.640 | -1.899 |
| | Backward + ID | **0.765** | 2.638 | -1.873 |
| Retweet | Forward | -6.405 | -5.625 | -0.780 |
| | Forward + ID | -6.370 | -5.602 | -0.767 |
| | Backward | -6.398 | -5.618 | -0.780 |
| | Backward + ID | **-6.367** | -5.600 | -0.767 |
| Taxi | Forward | 0.473 | 0.697 | -0.224 |
| | Forward + ID | 0.525 | 0.733 | -0.208 |
| | Backward | 0.477 | 0.705 | -0.228 |
| | Backward + ID | **0.528** | 0.738 | -0.209 |
| Taobao | Forward | 1.207 | 2.643 | -1.435 |
| | Forward + ID | **1.332** | 2.742 | -1.410 |
| | Backward | 1.215 | 2.648 | -1.432 |
| | Backward + ID | **1.332** | 2.742 | -1.410 |
| StackOverflow | Forward | -2.249 | -0.676 | -1.572 |
| | Forward + ID | -2.174 | -0.644 | -1.530 |
| | Backward | -2.225 | -0.679 | -1.547 |
| | Backward + ID | **-2.165** | -0.636 | -1.529 |
| Last.fm | Forward | **-0.463** | 1.309 | -1.772 |
| | Forward + ID | -0.477 | 1.302 | -1.779 |
| | Backward | -0.474 | 1.303 | -1.777 |
| | Backward + ID | -0.496 | 1.294 | -1.790 |
| MIMIC-II | Forward | 0.555 | 0.847 | -0.292 |
| | Forward + ID | **1.319** | 1.405 | -0.086 |
| | Backward | 0.322 | 0.601 | -0.279 |
| | Backward + ID | 1.231 | 1.345 | -0.114 |
| EHRSHOT (10%) | Forward | -3.885 | 0.105 | -3.990 |
| | Forward + ID | **-3.848** | -0.021 | -3.827 |
| | Backward | -4.571 | -0.432 | -4.139 |
| | Backward + ID | -4.684 | -0.641 | -4.043 |

### D.7 Model Calibration

To further probe the models, we evaluate the *calibration* of MTPPs, as proposed by Bosser and Taieb [2023]. Calibration has a different focus than log-likelihood-based or accuracy-based evaluation. Calibration instead describes how well the *uncertainty* in the model is reflective of the observed data. It is important to note, however, a model can achieve perfect calibration simply by predicting the marginal distribution. Better calibration therefore *does not* necessarily indicate better predictive performance — only better calibrated errors — and so should be taken in context with the performance under other metrics. We provide summarized statistics for both probabilistic calibration error (PCE) for time calibration and expected calibration error (ECE) for mark calibration in Table 15, and visualize the calibration curves in Figs. 12 and 13.

We see that, on the whole, MTPP models produce fairly well-calibrated predictions. IFTPP is the best calibrated of the models, this may be as a result of having parametric distributions for inter-arrival time (although IFTPP does fail on some datasets such as Retweet). The S2P2 is particularly well calibrated in time (PCE) among intensity-based methods, suggesting again that our S2P2 is capturing time dependencies better than other models. It is also the second-best calibrated on mark prediction (ECE) on average.

Table 15: Calibration results for the models and datasets tests.

(a) Probabilistic calibration error (PCE) for time calibration in percentage.

| Model | Probabilistic Calibration Error (PCE) ($\downarrow$) | | | | | | | | Avg. Ranking ($\downarrow$) |
|---|---|---|---|---|---|---|---|---|---|
| | Amazon | Retweet | Taxi | Taobao | StackOverflow | Last.fm | MIMIC-II | EHRSHOT | |
| RMTPP | 13.67 (0.03) | 7.93 (0.62) | 3.50 (0.03) | 0.22 (0.16) | 1.94 (0.10) | 1.56 (0.01) | 3.63 (0.37) | 12.60 (0.37) | 6.1 |
| SAHP | 12.04 (1.02) | 8.51 (1.86) | 2.52 (0.99) | 3.18 (0.21) | 1.50 (0.57) | 2.53 (1.86) | 2.28 (0.44) | 20.20 (1.09) | 5.3 |
| THP | 12.38 (0.05) | 5.68 (0.08) | 3.34 (0.02) | 6.36 (0.04) | 2.06 (0.11) | 1.02 (0.08) | **1.10** (0.06) | 13.46 (0.45) | 5.4 |
| IFTPP | **1.59** (0.09) | 23.85 (0.26) | **0.40** (0.10) | **1.61** (0.74) | **0.84** (0.34) | **0.46** (0.44) | 1.75 (0.33) | 16.58 (3.34) | **2.6** |
| MHP | 12.22 (0.04) | 4.89 (0.16) | 3.43 (0.05) | 8.77 (0.40) | 1.58 (0.13) | 1.25 (0.05) | 6.21 (0.18) | 15.24 (0.92) | 6.0 |
| NHP | 8.45 (0.28) | **0.20** (0.19) | 0.87 (0.50) | 7.40 (0.68) | 1.51 (0.11) | 4.70 (0.13) | 5.92 (0.14) | **7.70** (0.49) | 3.6 |
| AttNHP | 6.36 (0.63) | 2.09 (0.85) | 0.84 (0.27) | 3.08 (0.16) | 1.65 (0.24) | 1.43 (0.14) | 4.70 (0.33) | OOM | 3.7 |
| S2P2 (Ours) | 5.88 (0.17) | 0.44 (0.27) | 0.55 (0.33) | 2.07 (0.32) | 1.03 (0.15) | 1.38 (0.52) | 11.70 (0.68) | 12.06 (0.54) | 2.8 |

(b) Expected calibration error (ECE) for mark calibration in percentage.

| Model | Expected Calibration Error (ECE) ($\downarrow$) | | | | | | | | Avg. Ranking ($\downarrow$) |
|---|---|---|---|---|---|---|---|---|---|
| | Amazon | Retweet | Taxi | Taobao | StackOverflow | Last.fm | MIMIC-II | EHRSHOT | |
| RMTPP | 6.58 (0.15) | 3.99 (4.28) | 2.42 (0.16) | 1.89 (0.24) | 2.10 (0.27) | 2.47 (0.45) | 2.79 (0.43) | 8.47 (0.31) | 5.8 |
| SAHP | 8.17 (2.00) | 6.27 (2.23) | 6.77 (0.21) | 2.68 (0.35) | 1.71 (0.77) | 6.26 (4.30) | 5.41 (0.26) | 5.85 (1.95) | 6.8 |
| THP | 2.06 (0.17) | 1.26 (0.11) | 1.76 (0.07) | 6.51 (0.03) | **0.81** (0.14) | 3.42 (0.70) | 2.16 (0.39) | 8.95 (0.91) | 5.0 |
| IFTPP | **0.46** (0.10) | 0.95 (1.12) | **0.55** (0.19) | **1.20** (0.20) | 1.28 (0.54) | 0.66 (0.05) | **1.39** (0.23) | **1.99** (0.61) | **1.8** |
| MHP | 1.65 (0.16) | 1.18 (0.12) | 1.91 (0.11) | 4.15 (0.36) | 0.82 (0.18) | 2.83 (0.50) | 2.22 (0.24) | 10.00 (1.71) | 4.8 |
| NHP | 8.30 (0.21) | **0.35** (0.06) | 0.79 (0.10) | 5.59 (0.69) | 1.31 (0.16) | 3.41 (0.41) | 2.24 (0.32) | 4.18 (0.69) | 4.9 |
| AttNHP | 3.13 (0.61) | 0.52 (0.16) | 0.56 (0.10) | 2.47 (0.12) | 1.37 (0.42) | **0.61** (0.16) | 2.23 (0.50) | OOM | 3.4 |
| S2P2 (Ours) | 0.88 (0.34) | 0.52 (0.13) | 0.58 (0.12) | 1.96 (0.67) | 1.98 (0.19) | 1.01 (0.63) | 1.62 (0.24) | 2.51 (0.44) | 3.0 |

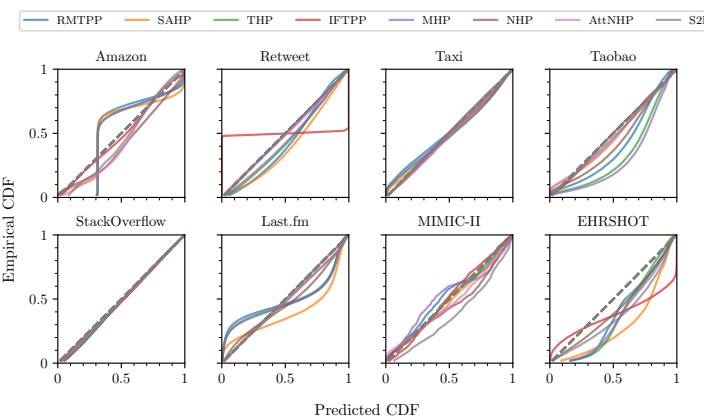

Figure 12: Reliability diagram for predicted inter-arrival time for each model on all datasets. Diagonal dashed lines refer to perfect calibration.

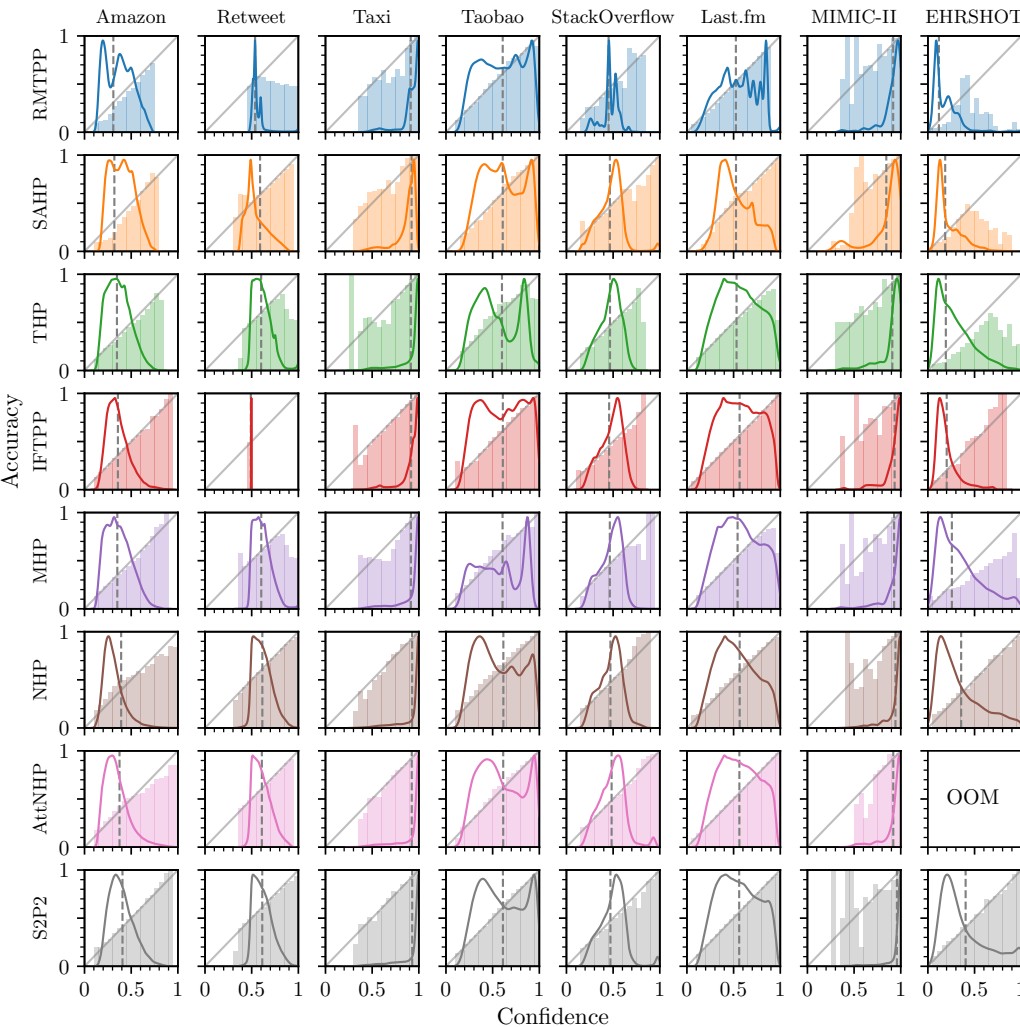

Figure 13: Reliability diagram for mark prediction of all models and all datasets. The $x$-axis specifies the confidence of model estimates grouped into 20 bins, and the $y$-axis of the bar plot is the model accuracy within that bin. The diagonal lines represent perfect calibration. The solid curves depict the distribution of confidences, and do not share the $y$-axis. The grey dashed lines indicate the overall prediction accuracy of the model for the next event conditioned on true event time.

Finally, in Figs. 14 and 15 we plot the log-likelihood of time and mark respectively, versus their corresponding calibration results, to provide an overall view of the performances of different models. Our S2P2 model consistently achieves higher log-likelihood while maintaining good calibration on both time and mark components on most datasets.

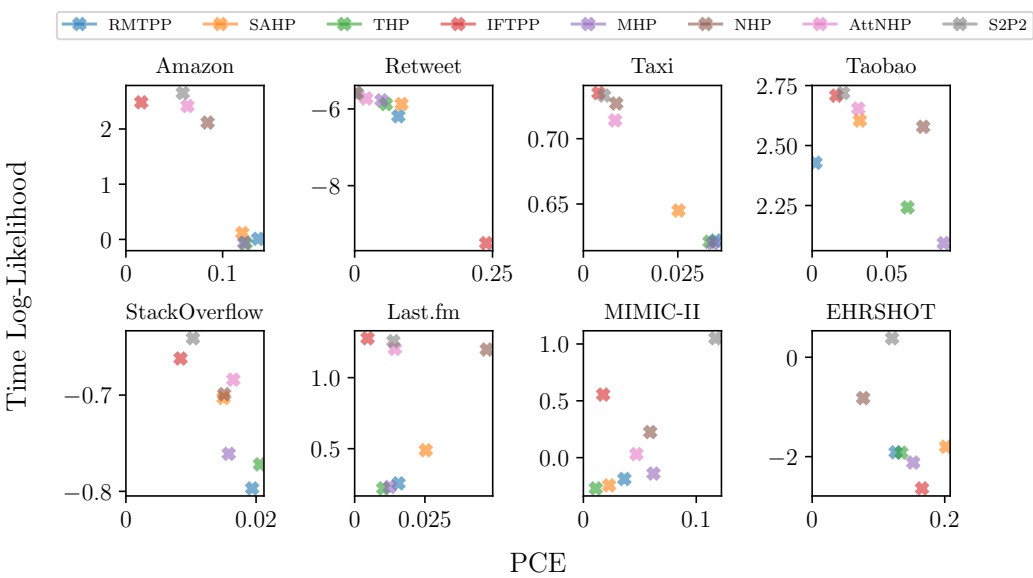

Figure 14: Log-likelihood of time vs. PCE for all models grouped by datasets. Higher log-likelihood and lower PCE are better (i.e., top left corner).

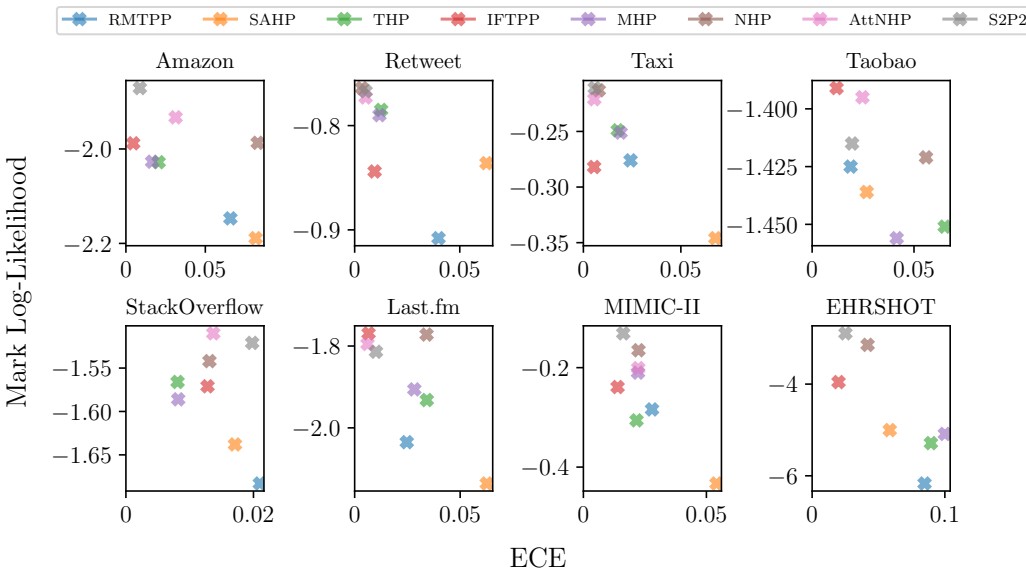

Figure 15: Log-likelihood of mark vs. ECE for all models grouped by datasets. Higher log-likelihood and lower ECE are better (i.e., top left corner).

