# OpenReview forum: "Deep Continuous-Time State-Space Models for Marked Event Sequences"
_NeurIPS.cc/2025/Conference — NeurIPS 2025 spotlight_

### Official Review · Reviewer_V2CN · 2025-06-17

**Clarity:** 3
**Significance:** 3
**Originality:** 3
**Rating:** 5
**Confidence:** 1

**Summary:**

This paper proposes S2P2, a deep learning architecture for modeling Marked Temporal Point Processes (MTPPs) by blending ideas from Hawkes processes and state space models (SSMs). The authors introduce a new recurrent building block, the Latent Linear Hawkes (LLH) layer, which generalizes the dynamics of linear Hawkes processes using the expressive, continuous-time latent recurrences of SSMs.
S2P2 is tested on snythetic data as well as eight real-world datasets.

**Questions:**

In practice, how does S2P2 handle sequences with >10,000 events? Are there any memory or stability concerns in such regimes, especially when evaluating intermediate intensities at random times?

**Ethical Concerns:**

["NO or VERY MINOR ethics concerns only"]

**Final Justification:**

I appreciate the author’s thoughtful responses to my questions. I keep the original score.

**Limitations:**

yes

**Paper Formatting Concerns:**

No concerns

**Quality:**

3

**Strengths And Weaknesses:**

I have to say, that I am not very familiar with models for temporal point processes.
Still I found the following strengths in the paper:

Rather than just using a black box model, their latent linear Hawkes layer has an inductive bias towards event sequences which extends the classical linear hawkes process.

The figures, especially figure 2, are highly informative and help the reader to understand the main ideas of the paper.

The experimental section includes synthetic data as well as real-world datasets and highlights the efficacy of the proposed method.

---

> ### Author Rebuttal · Authors · 2025-07-30
>
> We sincerely thank the reviewer for their positive evaluation and for highlighting the strengths in our model design, visualization, and experiments. We are glad that the inductive bias of the latent linear Hawkes layer and the clarity of the figures came through clearly, even for someone unfamiliar with the specific area.
>
> **How does S2P2 handle long sequences?**
>
> The memory usage for S2P2 scales linearly with the sequence length, enabling it to handle long sequences efficiently in practice. As shown in Section B.5 of the Appendix, we benchmarked inference latency for S2P2 and other models across sequences ranging from 8 to 500,000 events. _In these tests, S2P2 exhibited no memory or stability issues during inference._
>
> We expect this stability to extend to training on long sequences as well. This is due to the way the latent dynamics in the latent linear Hawkes layer are parameterized: the hidden states are guaranteed to converge to a steady state over time, which helps mitigate instability in long-range computations.
>
>
> **Summary**
>
> We thank the reviewer again for their positive and thoughtful comments. We’re glad that our explanations and figures helped make the core ideas accessible, and we hope our response clarifies S2P2’s scalability and stability on long sequences. Please don’t hesitate to reach out if we can provide any additional clarifications or details to assist your understanding further.
>
> -- Submission 20144 Authors

---

> ### Comment · Reviewer_V2CN · 2025-08-01
>
> I appreciate the author’s thoughtful responses to my questions. I keep the original score.

---

### Official Review · Reviewer_kua6 · 2025-06-30

**Clarity:** 3
**Significance:** 3
**Originality:** 3
**Rating:** 5
**Confidence:** 4

**Summary:**

The paper adds to a line of work on adding non-linear activation to the classical Hawkes process framework, in order to capture more complex marked point processes. The authors describe the model carefully, and explain the advantages in terms of expressivity and computational efficiency w.r.t the state of the art. Extensive experiments with synthetic and real-world data establish the power of the model.

**Questions:**

In the experiments, I did not understand how hyperparameters (# layers, layer width in particular) were chosen. How sensitive are results to these?

Details:
- what is ZOH? please define

**Ethical Concerns:**

["NO or VERY MINOR ethics concerns only"]

**Limitations:**

Yes

**Quality:**

4

**Strengths And Weaknesses:**

The paper is very carefully written in terms of defining and motivating the model, and was a pleasure to read - although I wonder whether they actually pack too much into this paper (thanks to unlimited supplementary).

The model is intuitive and well explained. The relationship with the state of the art is less so: it is difficult to appreciate the difference in assumptions/model expressivity with NHP, AttNHP, IFTPP in particular, and why S2P2 is more expressive. Again, I think the authors are forced to be too cursory given the desire to pack everything into this paper. From what I can judge, the model makes sense and has well-reasoned advantages, essentially combining well-known features of state-space models with specific computational tricks to make it work for (marked) point processes.

---

> ### Author Rebuttal · Authors · 2025-07-30
>
> We thank the reviewer for their thoughtful and constructive feedback, as well as their positive remarks on the clarity and motivation of our model. We are glad to hear the paper was a pleasure to read and that the core modeling ideas and empirical strengths came through. Below, we address your questions and suggestions in detail.
>
> **S2P2’s Relationship to State of the Art is Unclear**
>
> We intended to raise comparisons between S2P2 and other baselines at the relevant points throughout the paper.  However, as the reviewer points out, we did therefore forego a single explicit block describing the baseline methods (also partly due to space constraints). We will make the descriptions and distinctions between models explicit in the related works and experiments section in any camera-ready version.
>
> Please also let us know if there was anything specific on regarding baselines that you would like further clarified here!
>
>
> **On Hyperparameter Selection**
>
> Thank you for this question. Due to space, we deferred much of this discussion to the appendix. We will use part of the extra page in a camera ready paper to (a) bring up a high-level summary of our hyperparameter search from Appendix C to the main paper, and (b) make the pointer to Appendix C more visible in the main paper.
>
> To summarize here: S2P2’s main hyperparameters are: the number of layers $L$, the hidden state dimension $P$, and the residual stream dimension $H$. Intuitively, $P$ controls the model’s memory capacity, while $H$ limits the rank of the final transform from hidden states to marked intensities. For the experiments in Tables 1 and 2, individual to each baseline and dataset pairing, we performed a grid search over all hyperparameters to select configurations maximizing validation log-likelihood (note we report fully held-out test performance in the tables). The search space and best configurations are detailed in Appendix C.1 and Table 6.
>
> Regarding sensitivity, we generally found a range of reasonable hyperparameters to yield similar performance. We found that multiple layers $L$ was key for performance, but that there was not a critical dependence on depth. The dimensions $P$ and $H$ were more sensitive across datasets with a wider range, i.e., 16 to 128, being typically acceptable. A reasonable starting point is $H=P=32$ or $64$, followed by fine-tuning as needed when computational resources allow.
>
>
> **What is ZOH?**
>
> Thank you for spotting this! We accidentally cut _any_ mention of the full definition in Appendix B.2 (and Table 3) from the main text. We will add a short definition, discussion, and a reference to the full definition back to the main text.
>
> For completeness: ZOH stands for _zero-order hold_, a piecewise-constant interpolation technique applied to discrete-time signals to obtain continuous-time approximations. In our model, exact integration of inputs into the recurrent layers is computationally expensive (to the point of being intractable). Approximating the input with ZOH enables a tractable update rule that closely approximates the true continuous-time dynamics. The ZOH approximation is common in the deep SSM literature (see, e.g., S4D, S5).
>
> **Summary**
>
> We sincerely thank the reviewer for their careful and constructive feedback, as well as their kind words regarding the clarity and motivation of our work. We hope our detailed responses help clarify the points raised, and we welcome any further questions or requests for elaboration.
>
> -- Submission 20144 Authors

---

> > ### Comment · Reviewer_kua6 · 2025-08-06
> >
> > I thank the authors for the responses to my questions, I have no further issues with the paper and maintain my positive assessment.

---

### Official Review · Reviewer_CQ9o · 2025-07-05

**Clarity:** 3
**Significance:** 4
**Originality:** 4
**Rating:** 6
**Confidence:** 4

**Summary:**

the paper introduces S2P2 a MTPP model based on a neat connection between state-space models and linear Hawkes' processes. The key contribution of this paper is to utilize this connection to embed mark-specific dynamics in a marked temporal sequence to a, potentially more expressive, latent space. The paper also offers computationally tractable and efficient training of the resulting in a practical and quite novel approach to sequence modeling.

**Questions:**

Q1. Can the approach be extended to non-categorical marks? It would be interesting since some of the applications of Hawkes' processes have worked with spatio-temporal setting with non-categorical marks.

Q2. how sensitive is the model for potentially noisy/adversarial observations -- either in observed time or intensity. In many sensor settings this is a common occurance, and there have been some works that have improved the robustness of Hawkes' processes (e.g., Chakraborty et al., AAAI'25)

**Ethical Concerns:**

["NO or VERY MINOR ethics concerns only"]

**Final Justification:**

I am quite satisfied with the submitted version and the resulting discussions. I will retain my rating.

**Limitations:**

There is no discussion, but I do not see that as a limitation.

**Paper Formatting Concerns:**

None.

**Quality:**

4

**Strengths And Weaknesses:**

__Strengths:__

1. The paper makes a novel and interesting connection between linear Hawkes processes and deep state-space models.

2. proposes efficient update handling through diagonalization which is technically quite interesting.

3. experimental results over synthetic and real-world data with small and large mark space clearly demonstrates the superiority of s2p2.


__Weaknesses:__

1. The paper is fairly dense technically - particularly the combination of diagonalization and discretization stages.

2. latent mark space is quite non-interpretable in comparison to seen marks.

---

> ### Author Rebuttal · Authors · 2025-07-30
>
> We thank the reviewer for their strong support and thoughtful feedback. We are glad to hear the core contributions and empirical results were found novel and impactful. Please find our response to your questions and comments below.
>
> **Weakness 1 (Paper is Dense)**
>
> We agree that the math can be a little dense in places. As you say, space constraints meant some exposition was deferred to the appendix (see, for instance, B.2). In the camera-ready version, we commit to using the additional space to add more signposting, intuitive explanations, and clarifying examples to improve readability and accessibility for a broader audience.
>
> **Weakness 2 (Latent Space is Non-Interpretable)**
>
> This is a good observation. Indeed, compared to parametric Hawkes models, which have more immediate interpretability, our latent mark representations trade interpretability for expressiveness. This tradeoff is common between neural MTPP models in general; S2P2 is similar in interpretability to, for instance, neural Hawkes process. While we did initially note this in the conclusion, we will have a more expanded discussion of this and opportunities this presents for future research.  For instance, as also noted by reviewer cv8v, recent work on interpretability in SSMs could be extended to S2P2, which we believe is a promising direction for future work.
>
> **Question 1 (Non-Categorical Marks)**
>
> Yes, S2P2 can naturally be extended to non-categorical marks with architectural adjustments tailored to the structure of the mark space. For instance, in a spatial setting with marks $k \in \mathbb{R}^2$, one could:
> - Embed the continuous-valued mark into a dense vector at the input, similar to how categorical marks are handled;
> - At output, rather than computing separate intensities $\lambda_k(t)$ for each mark, model the marked intensity as $\lambda_k(t) := \lambda(t) \cdot p(k | t)$, where $\lambda(t)$ is the total intensity and $p(k | t)$ is the conditional mark density over $\mathbb{R}^2$ (for spatial).
>
> The total intensity $\lambda(t)$ can be computed as in the categorical case, while the conditional density $p(k | t)$ could be parameterized by a Gaussian, mixture model, or more flexible methods (e.g., normalizing flows), all conditioned on the latent state $\mathbf{h}(t)$. This allows S2P2’s continuous-time latent dynamics to fully inform both time and mark predictions. These extensions are a fascinating direction to explore and potentially very impactful. We will add this in the discussion section of the camera-ready version.
>
> **Question 2 (Sensitivity to Adversarial Observations)**
>
> Formal evaluation under adversarial or corrupted conditions is an open direction for S2P2. We evaluated calibration as part of our suite of metrics, finding that, in general, models are well-calibrated—but we acknowledge we evaluated this under 'normal' conditions. Extending this evaluation to adversarial or corrupted is therefore a very natural follow up. We thank the reviewer for pointing to Chakraborty _et al._ (AAAI’25) as a starting point, and will include discussion of this in the discussion/conclusion section.
>
> **Summary**
>
> We again thank the reviewer for taking the time to review our submission, for providing such positive and helpful feedback, and some great suggestions for follow-up works! We are more than happy to answer any further questions you may have!
>
> -- Submission 20144 Authors

---

> > ### Comment · Reviewer_CQ9o · 2025-08-06
> >
> > I thank the authors for their responses. I have a couple of follow-up requests:
> >
> > * Would it possible to present some results --even if it is on a smaller scale dataset-- the impact of corrupted data?
> >
> > * I would ask for a similar thing for non-categorical marks as well.
> >
> > The reason for these requests is that without these, it is not clear what the authors will be able to add (as promised) in the form of discussion when the paper is revised.

---

> > > ### Author Response · Authors · 2025-08-07
> > >
> > > Thank you for the follow-up. Given the timing, we’d like to briefly clarify how we plan to reflect these points in the camera-ready version.
> > >
> > > **Non-categorical marks**:
> > > While our original focus was on categorical marks, we agree this is an exciting direction for future work. Given the structural differences involved in modeling continuous mark spaces, our intention was to include a forward-looking proposal rather than a concrete contribution. We will clarify this in the revised version and ensure the discussion more clearly communicates its exploratory nature.
> > >
> > > **Corrupted data**:
> > > We will cite Chakraborty _et al._ (AAAI’25) and expand the discussion to acknowledge robustness to noise as an open area that warrants further investigation with our approach and MTPPs in general. Due to time and space constraints, we are not planning to include additional experiments, but we appreciate the reviewer’s suggestion and will ensure the discussion reflects it constructively.
> > >
> > > We hope this resolves any remaining concerns. We appreciate the reviewer’s engagement and suggestions, especially so close to the deadline.

---

### Official Review · Reviewer_cv8v · 2025-07-20

**Clarity:** 3
**Significance:** 3
**Originality:** 4
**Rating:** 5
**Confidence:** 3

**Summary:**

This paper presents a novel state-space model (SSM)-inspired framework, S2P2, for modeling marked temporal point processes (MTTPs). At the core of the framework is a newly proposed Latent Linear Hawkes (LLH) layer, which naturally integrates the linear Hawkes process with SSM principles. The authors further demonstrate how LLH can be effectively combined with techniques developed for modern deep SSMs (e.g., S4, S5, Mamba), enabling scalable, efficient, and tractable model training. In synthetic experiments, S2P2 successfully recovers the underlying signals where existing methods fall short. On real-world datasets, S2P2 achieves overall SOTA performance.

**Questions:**

Q1: For the "parametric decoding head" in THP and MHP, are you referring to following operation?
$$\lambda_k(t|\mathcal{H}_t) = \text{softplus}_k\left(\alpha_k \frac{t-t_j}{t_j} + \mathbf{w}_k^T \mathbf{h}(t_j) + b_k\right)$$
What is the fundamental difference between this operation and the final 'Projection & Softplus' layer?

Q2: Could S2P2 serve as a pretrained model, where the output of its final hidden layer (i.e., the layer preceding the 'Projection & Softplus' module) is used for downstream tasks such as clinical classification in EHRSHOT? Given its expressiveness, S2P2's representations are expected to outperform the count-based baselines on the EHRSHOT leaderboard.

Q3: I am surprised that NHP seems to be significantly better than THP, as NHP was the baseline in THP. Did you make any modification to the original NHP?

**Ethical Concerns:**

["NO or VERY MINOR ethics concerns only"]

**Final Justification:**

My review was favorable and the authors' responses have further strengthened my assessment.

**Limitations:**

- Althrough you do not have to address the interpretability issue in this paper, it is worthwhile explore some frameworks that make SSMs more explanable in the discussion.
- Minor Fixes:
	- Eqn (3) no subscript in $\boldsymbol\alpha$ on the left hand side
	- line 211, should use `\mathcal`

**Paper Formatting Concerns:**

There is no formatting concern. The main content is within 9-page limit. Checklist is included.

**Quality:**

3

**Strengths And Weaknesses:**

Strengths:
- I really enjoyed reading this paper. It is well-organized, well-written, and easy to follow.
- The design of the LLH layer is clearly articulated, with a detailed explanation of how it combines LHP with SSM. The authors provide well-supported insights from both theoretical and engineering perspectives.
- The authors provide a complet set of experiments for both synthetic data and real-world datasets and suffcient baseline comparisons.

Weaknesses:
- For transformer-based HP models, analyzing their attention maps provides critical insights into how historical events influence the future. SSMs in general have weaker interpretability. However, given that S2P2 captures input-dependent dynamics, can these be visualized similarly to attention maps? Maybe there exist ways to calculate some implicit attention scores in a SSM (e.g. [1]) (You don't have to address this in for the rebuttal).

[1] Ali A, Zimerman I, Wolf L. The hidden attention of mamba models. arXiv preprint arXiv:2403.01590. 2024 Mar 3.

---

> ### Author Rebuttal · Authors · 2025-07-30
>
> We thank the reviewer for their thoughtful feedback and kind words about our paper. We appreciate the recognition of our theoretical framing, empirical results, and clarity. Below, we address each question and suggestion in turn.
>
>
> **On Model Interpretability**
>
> This is a great point; incorporating or extracting interpretability from S2P2 is an exciting direction for future work. In relation to the referenced paper: Conditioned on the event times, S2P2 computes update matrices similar in form to Mamba, hence S2P2 is eligible to utilize the method in [1] (and likely other techniques being developed for deep SSMs in general). For [1] specifically, it is worth noting that since S2P2 inherently lives in continuous time, it will exhibit a continuously varying form of this implicit attention. Fully taking advantage of this aspect will require special care, but, excitingly, would provide richer insights when compared to discrete-time attention maps.
>
> [1] Ali A, Zimerman I, Wolf L. The hidden attention of mamba models. arXiv preprint arXiv:2403.01590. 2024 Mar 3.
>
> **Question 1 (THP/MHP Decoding Head)**
>
> Yes, you are correct; this is the parametric decoding head that we referred to. We describe it as “parametric” because, after conditioning on a history $t_1, …, t_j$, the model encodes this sequence into a single hidden state $\mathbf{h}(t_j)$. The intensity function is then computed as
> $$
> \lambda_k(t|\mathcal{H}_t) = \text{softplus}_k\left(\alpha_k\frac{t-t_j}{t_j} + \mathbf{w}_k^\top\mathbf{h}(t_j) + b_k\right),
> $$
>  If we combine terms, conditioned on a particular history, the intensity for each mark simplifies to $\text{softplus}(a t + b)$, where $a,b$ are scalar values representing combinations of parameters/values from the initial expression.
>
> In contrast, S2P2 models the intensity as:
> $$
> \lambda_k(t | \mathcal{H}_t) = \text{softplus}_k\left(\mathbf{w}_k^\top \mathbf{h}(t) + b_k \right).
> $$
> We see now that a constituent of the input to the softplus is $\mathbf{h}(t)$ (instead of $\mathbf{h}(t_j)$).  As a result, there is no simplification rendering the intensity as the softplus of a linear-in-$t$ function (as before); it is instead the softplus of a _non-linear_ and _history-dependent_ function of $t$, because $h$ is defined by the entire S2P2 stack and event history. This is why we describe THP as being “parametric”; and why S2P2 can represent a much richer class of intensities across $t$. We will add further clarification of this in the methods section and appendix.
>
>
> **Question 2 (S2P2 as a Pretrained Backbone)**
>
> Yes, S2P2 can indeed be used as a pretrained model for downstream tasks. One can use the final hidden state $\mathbf{h}(t_n)$ after conditioning on $n$ events, or alternatively, leverage the latent state trajectory $\mathbf{h}(t)$ across time $t \geq t_n$ for richer representations. Given the expressiveness of S2P2, we agree this direction is promising. Exploring these variants systematically, e.g., via ablations on clinical classification in EHRSHOT, will be a valuable extension for future work. We will add this in the discussion section of any camera-ready version.
>
> **Question 3 (Empirical Performance of NHP vs. THP)**
>
> All baselines used up-to-date PyTorch implementations provided by the EasyTPP library [2]. The library’s implementations for NHP and THP are adopted from Yang _et al._ [3]. These versions were explicitly confirmed by the original authors as being correct (see footnote on pg. 8 of [3]). As noted in [3] and confirmed in our experiments, NHP often outperforms THP under this setup. We will clarify these implementation details in the final version.
>
> [2] Xue, Siqiao, et al. "EasyTPP: Towards Open Benchmarking Temporal Point Processes." _Proceedings of the Twelfth International Conference on Learning Representations (ICLR)_. 2024.
>
> [3] Yang, Chenghao, Hongyuan Mei, and Jason Eisner. "Transformer Embeddings of Irregularly Spaced Events and Their Participants." _Proceedings of the Tenth International Conference on Learning Representations (ICLR)_. 2022.
>
> **Minor Fixes**
>
> Thank you for spotting these typos! We will be sure to correct them in a camera-ready version.
>
>
> **Summary**
>
> We sincerely thank the reviewer for taking the time to assess our submission, for the encouraging and insightful feedback, and for pointing out areas where our presentation can be refined. We are happy to clarify anything further!
>
> -- Submission 20144 Authors

---

> > ### Comment · Reviewer_cv8v · 2025-08-04
> >
> > Thanks for these helpful responses. I will keep my score.

---

### Comment · Area_Chair_yPWA · 2025-08-01

Dear Reviewers,

The authors have posted their rebuttals. If you have any additional questions or require clarification, please add your comments as soon as possible—the author-reviewer discussion period ends in one week.

Thank you for your prompt attention.

Best regards,

AC

---

> ### Comment · Area_Chair_yPWA · 2025-08-06
>
> Here is a gentle reminder: The Author-Reviewer discussions will end by Aug 8, 11.59pm AoE. Please also be aware that there is a "Mandatory Acknowledgement" for the reviewers.

---

### Note · Authors · 2025-08-12

We thank the reviewers for their thorough evaluation of our work, and for their kind words and positive feedback on our submission.  We appreciate the discussion, and hope our feedback allayed the few queries there were.  All edits/updates discussed will be made in any camera ready version.

— Submission 20144 authors.

---

### Decision · Program_Chairs · 2025-09-17

**Decision:**

Accept (spotlight)

**Comment:**

The paper introduces **S2P2**, a state-space point process model that integrates linear Hawkes processes with modern deep state-space modeling techniques to effectively capture the dynamics of marked temporal point processes. Its key contributions are the Latent Linear Hawkes layer, efficient parallelizable inference, and strong empirical gains (33% average improvement over baselines across eight datasets). Reviewers highlighted the paper’s technical novelty, careful exposition, and thorough experiments as major strengths. Concerns raised were mainly about interpretability, clarity on comparisons with some baselines, dense mathematical exposition, and extensions to non-categorical marks or robustness to corrupted data. The rebuttal convincingly addressed these points by clarifying distinctions from prior work, elaborating on hyperparameter selection, scalability, and stability, and outlining interpretability and robustness as promising directions for future research. All reviewers maintained positive scores after discussion, with one strong accept. Given the combination of conceptual novelty, rigorous methodology, and state-of-the-art empirical performance, I recommend **acceptance**. This paper makes a significant and timely contribution to probabilistic modeling of event sequences and will likely have strong impact in both theoretical and applied domains.